# GNOTHI SEAUTON: EMPOWERING FAITHFUL SELF-INTERPRETABILITY IN BLACK-BOX TRANSFORMERS

**Shaobo Wang**[1,2]  **Hongxuan Tang**[2]  **Mingyang Wang**[2]  **Hongrui Zhang**[2]
**Xuyang Liu**[2,3]  **Weiya Li**[4]  **Xuming Hu**[5]  **Linfeng Zhang**[1,2*]

[1]School of Artificial Intelligence, Shanghai Jiao Tong University
[2]Efficient and Precision Intelligent Computing Lab, Shanghai Jiao Tong University
[3]Sichuan University    [4]Big Data and AI Lab, ICBC
[5]Hong Kong University of Science and Technology, Guangzhou
{shaobowang1009,zhanglinfeng}@sjtu.edu.cn

## ABSTRACT

The debate between self-interpretable models and post-hoc explanations for black-box models is central to Explainable AI (XAI). Self-interpretable models, such as concept-based networks, offer insights by connecting decisions to human-understandable concepts but often struggle with performance and scalability. Conversely, post-hoc methods like Shapley values, while theoretically robust, are computationally expensive and resource-intensive. To bridge the gap between these two lines of research, we propose a novel method that combines their strengths, providing theoretically guaranteed self-interpretability for black-box models without compromising prediction accuracy. Specifically, we introduce a parameter-efficient pipeline, *AutoGnothi*, which integrates a small side network into the black-box model, allowing it to generate Shapley value explanations without changing the original network parameters. This side-tuning approach significantly reduces memory, training, and inference costs, outperforming traditional parameter-efficient methods, where full fine-tuning serves as the optimal baseline. *AutoGnothi* enables the black-box model to predict and explain its predictions with minimal overhead. Extensive experiments show that *AutoGnothi* offers accurate explanations for both vision and language tasks, delivering superior computational efficiency with comparable interpretability.

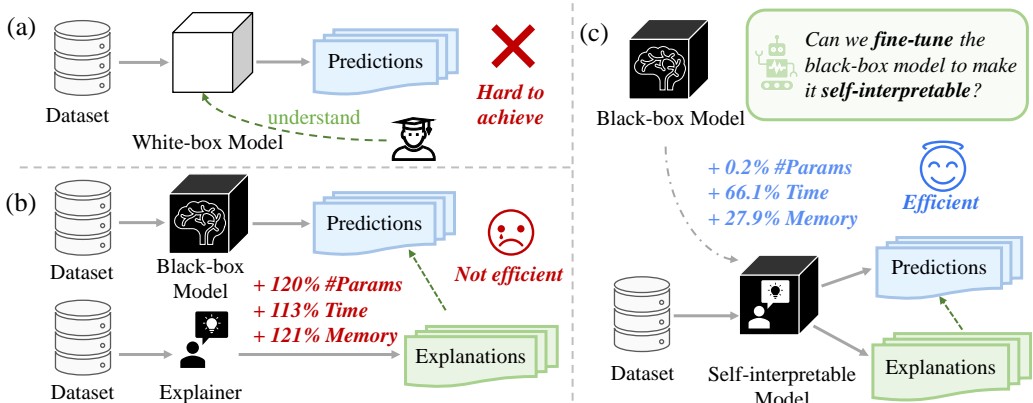

Figure 1: **Different paradigms towards XAI.** (a) The ideal paradigm for XAI envisions using white-box models for prediction, which are inherently self-interpretable by design but hard to achieve. (b) The previous paradigm involves post-hoc explanations of black-box models by training a separate, heavy-weight explainer. (c) We propose a novel parameter-efficient paradigm, *AutoGnothi*, which fine-tunes the black-box model to make it self-interpretable.

---

*Corresponding Author.

## 1 INTRODUCTION

Explainable AI (XAI) has gained increasing significance as AI systems are widely deployed in both vision (Dosovitskiy, 2020; Radford et al., 2021; Kirillov et al., 2023) and language domains (Devlin et al., 2019; Brown, 2020; Achiam et al., 2023). Ensuring interpretability in these systems is vital for fostering trust, ensuring fairness, and adhering to legal standards, particularly for complex models such as transformers. As illustrated in Figure 1(a), the ideal paradigm for XAI involves designing inherently transparent models that deliver superior performance. However, given the challenges in achieving this, current XAI methodologies can be broadly classified into two main categories: developing *self-interpretable models* and providing *post-hoc explanations* for black-box models.

**Designing Self-Interpretable Models:** Several notable efforts have focused on designing self-interpretable models that are grounded in solid mathematical foundations or learned concepts. Among these, concept-based networks have emerged as a representative approach linking model decisions to predefined, human-understandable concepts (Kim et al., 2018; Koh et al., 2020; Alvarez-Melis & Jaakkola, 2018). However, incorporating hand-crafted concepts often introduces a trade-off between interpretability and performance, as these models typically compromise the performance of the primary task. Moreover, such methods are often closely tied to specific architectures, which limits their scalability and transferability to other tasks. Furthermore, the explanations generated by concept-based models often lack a rigorous theoretical foundation, raising concerns about their reliability and overall trustworthiness.

**Explaining Black-Box Models:** Given the challenges of designing self-interpretable models for practical applications, post-hoc explanations for black-box models have become a widely adopted alternative. Among these, Shapley value-based methods (Shapley, 1953) are particularly valued for their theoretical rigor and adherence to four principled axioms (Young, 1985). However, calculating exact Shapley values involves evaluating all possible feature combinations, which scales exponentially with the number of features, making direct computation impractical for models with high-dimensional inputs. To alleviate this, methods like Fast-SHAP (Jethani et al., 2021) and ViT-Shapley (Covert et al., 2022) employ proxy explainers that estimate Shapley values during inference, significantly reducing the number of evaluations needed. While these approaches reduce some computational costs, training a separate explainer remains resource-intensive.

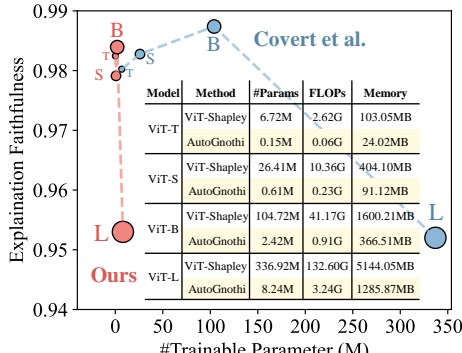

Figure 2: **Explanation quality on the ImageNette dataset using different ViTs.** Our *AutoGnothi* significantly reduces the number of trainable parameters, computational costs (FLOPs), and training GPU memory storage without compromising explanation quality.

For example, training a Vision Transformer (ViT) explainer requires more than twice the training GPU memory compared to the ViT classifier itself. Moreover, solely depending on post-hoc explanations for black-box models is not ideal in high-stakes decision-making scenarios, where immediate and reliable interpretability is required (Rudin, 2019).

To bridge the gap between existing methods and address the aforementioned challenges, the core objective of our research is to *achieve theoretically guaranteed self-interpretability in advanced neural networks without sacrificing prediction performance, while minimizing training, memory, and inference costs*. To this end, we propose a novel paradigm, *AutoGnothi*, which leverages parameter-efficient transfer learning (PETL) to substantially reduce the high training, memory, and inference costs associated with obtaining explainers. As depicted in Figure 3(a), traditional model-specific methods require two training stages: (i) fine-tuning a pre-trained model into a surrogate model, and (ii) training an explainer using the surrogate model. During inference, the original model is used for prediction, while a separate explainer network generates explanations, leading to two inference passes and double the storage overhead. In contrast, *AutoGnothi* utilizes side-tuning to reduce both training and memory costs, as shown in Figure 3(b). By incorporating an additive side branch parallel to the pre-trained model, we efficiently obtain a surrogate side network through side-tuning. We then apply the same strategy to develop the explainer side network, enabling simultaneous predic-

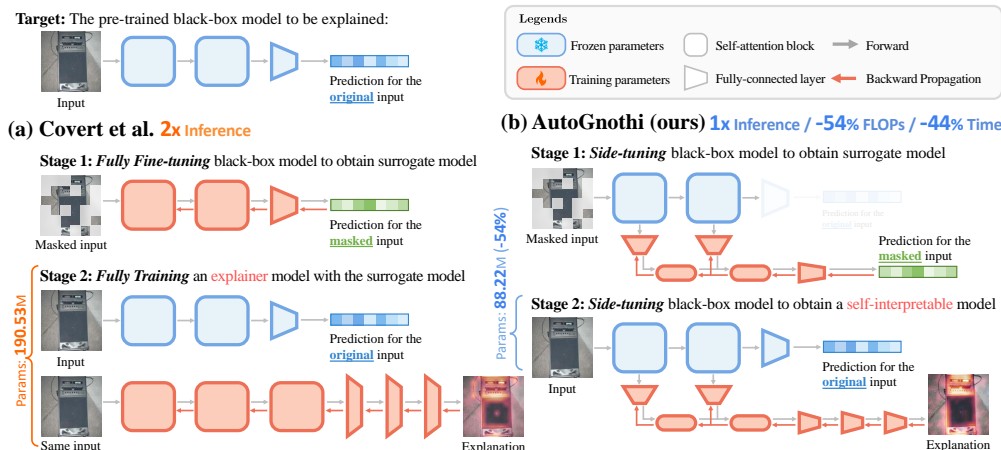

Figure 3: **Overview of *AutoGnothi* compared to prior work.** (a) ViT-Shapley (Covert et al., 2022) fully fine-tunes the black-box model to create a surrogate model, then trains a separate explainer based on the surrogate, which is resource-intensive. (b) We employ side-tuning to efficiently obtain both the black-box model and explainer, significantly reducing training costs. *AutoGnothi* uses a single model to simultaneously generate predictions and explanations, lowering inference costs by leveraging shared features. In contrast, ViT-Shapley needs to load two models for prediction and explanation, respectively, and infers two times. *AutoGnothi* enables self-interpretability for an arbitrary black-box model. We ignore the positional encoding associated with the pipeline.

tion and explanation in a single inference step. An illustrative example comparing the efficiency of *AutoGnothi* with previous methods is presented in Figure 2.

More importantly, *AutoGnothi* achieves self-interpretability without compromising prediction accuracy. Unlike a simple application of PETL, where full fine-tuning is considered the optimal baseline, our approach goes further. Experimental results show that relying on full fine-tuning to achieve self-interpretability often leads to degraded performance in either prediction or explanation tasks. In contrast, *AutoGnothi* maintains prediction accuracy while achieving self-interpretability by leveraging the intrinsic correlation between prediction and explanation. Beyond classical PETL, which primarily focuses on training efficiency, *AutoGnothi* also enhances inference efficiency through self-interpretation while keeping its faithfulness (see Section 4.2 for further discussion). Our key contributions are as follows:

1. **Efficient Explanation**: We introduce a novel PETL pipeline, *AutoGnothi*, which enables any black-box models, *e.g.*, transformers, to become self-interpretable without affecting the original task parameters. By integrating and fine-tuning an additive side network, suprisingly surpasses previous methods in training, inference, and memory efficiency.

2. **Self-interpretability**: We achieve theoretically guaranteed self-interpretability for the black-box model with the Shapley value, without any influence on the original model's prediction accuracy.

3. **Broad Applications on both Vision and Language Models**: We conducted experiments on the most widely used models, including ViT (Dosovitskiy, 2020) for image classification and BERT (Devlin et al., 2019) for sentimental analysis, showing that our methods outperform well on explanation quality. Specifically, for the ViT-base model pre-trained on ImageNette, our surrogates achieve a $97\%$ reduction in trainable parameters and $72\%$ reduction in training memory with comparable accuracy. For explainers, we achieve a $98\%$ reduction in trainable parameters, $77\%$ reduction in training memory. For generating explanation, *AutoGnothi* achieves $54\%$ reduction in inference computation, $44\%$ reduction in inference time, and a total parameter reduction of $54\%$.

## 2 RELATED WORK

**Explaining Black-Box Models with Shapley Values:** Among post-hoc explanation methods, the Shapley value (Shapley, 1953) is widely recognized as a faithful and theoretically sound metric for feature attribution, uniquely satisfying four key axioms: *efficiency*, *symmetry*, *linearity*, and *dummy*. However, computing Shapley values is computationally expensive, requiring $\mathcal{O}(2^n)$ operations to calculate a single Shapley value for one feature in a set of size $n$. To alleviate this computational burden, various approaches have been proposed to expedite Shapley value computation, which can

be broadly divided into *model-agnostic* and *model-specific* methods (Chen et al., 2023a). Model-agnostic techniques, such as KernelSHAP (Lundberg, 2017) and its enhancements (Covert & Lee, 2020), approximate Shapley values by sampling subsets of feature combinations. Nevertheless, when the feature set is large, the sampling cost remains prohibitive, and reducing this cost compromises the accuracy of the explanations, as fewer samples lead to less reliable estimates of feature importance. Conversely, model-specific methods, such as FastSHAP (Jethani et al., 2021) and ViT-Shapley (Covert et al., 2022), employ a trained proxy explainer to accelerate the estimation during inference, though these methods still involve significant training costs to develop the explainer.

**Parameter Efficient Transfer Learning (PETL):** PETL aims to achieve the performance of full fine-tuning while significantly reducing training costs by updating only a small subset of parameters. In this context, Adapters (Houlsby et al., 2019; Chen et al., 2023b) introduce trainable bottleneck modules into transformer layers, enabling models to deliver competitive results with minimal parameter adjustments. Another widely adopted method, LoRA (Hu et al., 2021), applies low-rank decomposition to the attention layer weights. Our work aligns more closely with side-tuning methods, where Side-Tuning (Zhang et al., 2020) integrates an auxiliary network that merges its representations with the backbone at the final layer, demonstrating effectiveness across diverse tasks in models like ResNet and BERT. LST (Sung et al., 2022) further improves this approach by reducing memory consumption through a ladder side network design. However, none of these methods explore the transfer of interpretability from the main model to the side network, leaving this a largely unexplored area in side-tuning and PETL research.

## 3 BACKGROUND

### 3.1 SHAPLEY VALUES

The Shapley value, originally introduced in game theory (Shapley, 1953), provides a method to fairly distribute rewards among players in coalitional games. In this framework, a set function assigns a value to any subset of players, corresponding to the reward earned by that subset. In machine learning scenarios, input variables are typically regarded as players, and a deep neural network (DNN) serves as the value function, assigning importance (saliency) to each input variable.

Let $s \in \{0, 1\}^d$ be an indicator vector representing a specific variable subset for a sample $x = [x_1, x_2, \ldots, x_d]^\top \in \mathbb{R}^d$. Specifically, $x_s$ denotes the variables indicated by $s$, while those not in $s$ are replaced by a masked value (*e.g.*, a baseline value). Let $e_i \in \mathbb{R}^d$ denote the vector with a one in the $i$-th position and zeros elsewhere. For a game involving $d$ players—or equivalently, a DNN $v : \{0, 1\}^d \to \mathbb{R}$ with $d$ input variables—the Shapley values are denoted by $\phi_v(x_1), \ldots, \phi_v(x_d)$. Each $\phi_v(x_i) \in \mathbb{R}$ represents the value attributed to the $i$-th input variable $x_i$ in the sample $x$. The Shapley value $\phi_v(x_i)$ is computed as follows:

$$\phi_v(x_i) = \frac{1}{d} \sum_{s:s_i=0} \binom{d-1}{\mathbf{1}^\top s} (v(x_{s+e_i}) - v(x_s)). \tag{1}$$

Intuitively, Eq. (1) captures the average marginal contribution of the $i$-th player to the overall reward by considering all possible subsets in which player $i$ could be included. Shapley values satisfy four key axioms: *linearity, dummy player, symmetry*, and *efficiency* (Young, 1985). These axioms ensure a fair and consistent distribution of the total reward among all players.

### 3.2 MODEL-BASED ESTIMATION OF SHAPLEY VALUES

Calculating Shapley values to explain individual predictions presents substantial computational challenges (Chen et al., 2023a). To mitigate this burden, these values are typically approximated using sampling-based estimators, such as those in (Lundberg, 2017; Covert & Lee, 2020), though the sampling cost remains considerable. Recently, a more efficient model-based approach, introduced in (Jethani et al., 2021), accelerates the approximation by training a proxy explainer to compute Shapley values through a single model inference. However, this method has not been validated on advanced neural architectures such as transformers.

Building on this, ViT Shapley (Covert et al., 2022) was introduced to train a ViT explainer that interprets a pre-trained ViT model $f$. As shown in Figure 3(a), the learning process of the explainer

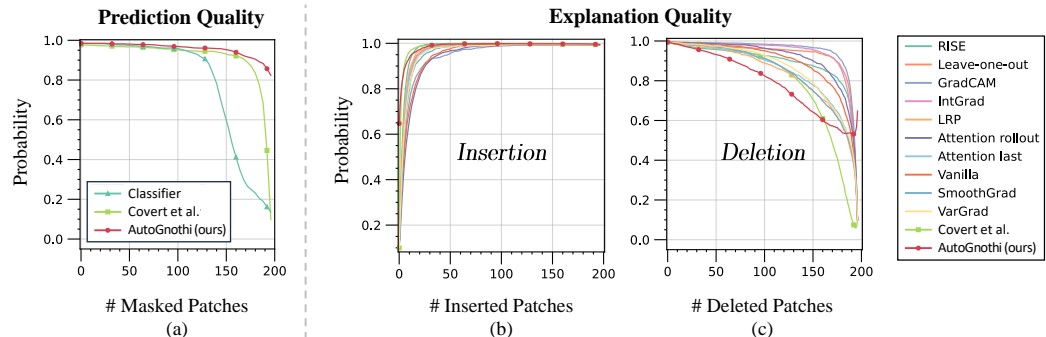

Figure 4: **Training performance of surrogate and explainer models.** (a) Prediction accuracy of masked inputs for the original classifier, the surrogate model trained with ViT Shapley (Covert et al., 2022), and our *AutoGnothi*. *AutoGnothi* shows greater robustness as the number of masked patches increases. For each mask size, we randomly sampled 100 images and generated 10 random masks. The curve represents the average prediction probability. (b) Explanation quality, measured by insertion and deletion metrics, for various explanation methods. We randomly sampled 1,000 images and averaged the prediction probabilities to assess insertion and deletion performance. All experiments were conducted on the ImageNette dataset using the ViT-base model.

consists of two stages. In stage 1, a surrogate model $g_\beta$ with parameters $\beta$ is generated by fine-tuning the pre-trained ViT classifier $f$ to handle partial information, which is used for calculating the masked variables in the Shapley value. This involves aligning the output distributions of the surrogate model $g_\beta$ with the classifier $f$. The surrogate is optimized with the following objective:

$$\mathcal{L}_{\text{surr}}(\beta) = \mathbb{E}_{x \sim p(x), s \sim \mathcal{U}(0,d)} \left[ D_{\text{KL}} \left( f(x) \| g_\beta(x_s) \right) \right], \tag{2}$$

where $\mathcal{U}(0,d)$ is a uniform distribution for sampling $s$. Then, in stage 2, an explainer $\phi_\theta$ with parameters $\theta$ is trained to generate explanations of the predictions of the black-box ViT $f$, utilizing the surrogate model $g_\beta$. This optimization method was first proposed by (Charnes et al., 1988) and later applied in (Lundberg, 2017; Jethani et al., 2021; Covert et al., 2022). Let $p(x)$ and $p(x,y)$ denote the distributions of input and input-label pairs, respectively. Specifically, the loss for training the explainer is:

$$\mathcal{L}_{\text{exp}}(\theta) = \mathbb{E}_{(x,y) \sim p(x,y), s \sim q(s)} \left[ \left( g_\beta(x_s|y) - g_\beta(x_0|y) - s^\top \phi_\theta(x,y) \right)^2 \right] \tag{3}$$

$$\text{s.t.} \quad \mathbf{1}^\top \phi_\theta(x,y) = g_\beta(x_1|y) - g_\beta(x_0|y) \quad \forall (x,y), \tag{Efficiency}$$

where the constraint in the loss function is enforced to satisfy the efficiency axiom of the Shapley value, and $q(s)$ is defined with the Shapley kernel (Charnes et al., 1988) as follows:

$$q(s) \propto \frac{d-1}{\binom{d}{\mathbf{1}^\top s}(\mathbf{1}^\top s)(d - \mathbf{1}^\top s)} \quad \forall s : 0 < \mathbf{1}^\top s < d, \tag{Shapley Kernel}$$

In addition, the ViT explainer $\phi_\theta$ has the same number of multi-head self-attention (MSA) layers as the feature backbone, and includes three additional MSA layers and a fully-connected (FC) layer as the explanation head. By learning the explainer $\phi_\theta$ to estimate Shapley values, the computational cost is reduced to a constant complexity of $\mathcal{O}(1)$.

## 4 METHOD

### 4.1 EFFICIENTLY TRAINING THE SHAPLEY VALUE EXPLAINER FOR BLACK-BOX MODELS

As discussed in Section 3.2, existing methods for approximating Shapley values require two stages. First, the black-box model $f$ is fully fine-tuned to obtain a surrogate model $g$ with the same trainable parameters and memory cost as $f$. Then, an explainer $\phi$ is fully trained using $g$. These stages at least double the training, memory, and inference costs compared to using the black-box model $f$ alone, making these methods impractical for large models.

To improve training, memory, and inference efficiency, we propose a side-tuning pipeline called *AutoGnothi*, as shown in Figure 3(b). Building on ideas from PETL, we adapt Ladder Side-Tuning (LST) (Sung et al., 2022) by incorporating an additive side network. This side network

Table 1: Comparison of training and memory efficiency across different models. We evaluated memory consumption and trainable parameters for previous methods (Covert et al., 2022) and *AutoGnothi* across a range of models and tasks. For surrogate models, we measured classification accuracy, while for explainers, we assessed explanation quality using insertion and deletion metrics. It is important to note that prediction accuracy for the classifier is evaluated on normal inputs, whereas for the surrogate model, accuracy is measured on masked inputs.

| Dataset | | ImageNette | | | | MURA | Pet | Yelp |
|---|---|---|---|---|---|---|---|---|
| Model | | ViT-T | ViT-S | ViT-B | ViT-L | ViT-B | ViT-B | BERT-B |
| Classifier to be explained | Memory (MB) | 84.90 | 331.80 | 1311.61 | 4631.25 | 1311.50 | 1311.93 | 1676.59 |
| | #Params (M) | 5.53 | 21.67 | 85.81 | 303.31 | 85.80 | 85.83 | 109.48 |
| | Accuracy ($\uparrow$) | 0.9791 | 0.9944 | 0.9944 | 0.9964 | 0.8186 | 0.9469 | 0.9010 |
| Surrogate (Covert et al.) | Memory (MB) | 84.90 | 331.80 | 1311.61 | 4631.25 | 1311.50 | 1311.93 | 1676.59 |
| | #Params (M) | 5.53 | 21.67 | 85.81 | 303.31 | 85.80 | 85.83 | 109.48 |
| | Accuracy ($\uparrow$) | 0.9822 | 0.9934 | 0.9939 | 0.9975 | 0.8233 | 0.9469 | 0.9490 |
| Surrogate (*AutoGnothi*) | Memory (MB) | 23.83 (-72%) | 92.38 (-72%) | 363.64 (-72%) | 1280.80 (-72%) | 438.10 (-67%) | 363.76 (-72%) | 532.71 (-68%) |
| | #Params (M) | 0.14 (-97%) | 0.56 (-97%) | 2.23 (-97%) | 7.91 (-97%) | 7.11 (-92%) | 2.23 (-97%) | 7.15 (-93%) |
| | Accuracy ($\uparrow$) | 0.9791 | 0.9939 | 0.9959 | 0.9959 | 0.8139 | 0.9422 | 0.9280 |
| Explainer (Covert et al.) | Memory (MB) | 103.05 | 404.10 | 1600.21 | 5144.05 | 1599.79 | 1601.47 | 1955.87 |
| | #Params (M) | 6.72 | 26.41 | 104.72 | 336.92 | 104.69 | 104.80 | 127.79 |
| | Insertion ($\uparrow$) | 0.9824 | 0.9828 | 0.9839 | 0.9843 | 0.9319 | 0.9422 | 0.9620 |
| | Deletion ($\downarrow$) | 0.5243 | 0.6865 | 0.8121 | 0.7646 | 0.4199 | 0.4958 | 0.1725 |
| Explainer (*AutoGnothi*) | Memory (MB) | 24.02 (-77%) | 93.12 (-77%) | 366.51 (-77%) | 1285.87 (-75%) | 449.38 (-72%) | 366.75 (-77%) | 685.32 (-65%) |
| | #Params (M) | 0.15 (-98%) | 0.61 (-98%) | 2.42 (-98%) | 8.24 (-98%) | 7.85 (-93%) | 2.43 (-98%) | 17.15 (-87%) |
| | Insertion ($\uparrow$) | 0.9802 | 0.9791 | 0.9874 | 0.9837 | 0.9292 | 0.9384 | 0.9588 |
| | Deletion ($\downarrow$) | 0.5097 | 0.6667 | 0.7954 | 0.6570 | 0.4116 | 0.4888 | 0.1004 |

separates the trainable parameters from the backbone model $f$ and adapts the model to a different task. It is a lightweight version of $f$, with weights and hidden state dimensions scaled by a factor of $1/r$, where $r$ is a reduction factor (*e.g.*, $r = 4$ or $8$). For instance, if the backbone $f$ has a 768-dimensional hidden state, then with $r = 8$, the side network has a hidden state dimension of 96. By computing gradients solely for the side network, this design avoids a backward pass through the main backbone, improving training and memory efficiency. The formulation combines the frozen pre-trained backbone and the side-tuner with learnable parameters $\beta$ as:

$$y^{\text{main}} = \underbrace{f}_{\text{frozen}}(x), \quad y^{\text{surr}} = \underbrace{g_\beta}_{\text{trainable}}(x_s), \quad \phi^{\text{exp}} = \underbrace{\phi_\theta}_{\text{trainable}}(x), \tag{4}$$

where $g_\beta$ is trained by minimizing the loss in Eq.(2), and $\phi_\theta$ is trained by minimizing the loss in Eq.(3), respectively.

### 4.1.1 OBTAINING THE SURROGATE

To obtain the surrogate model, *AutoGnothi* applies LST directly to the black-box model $f$, utilizing the additive side branch $g$ with parameters $\beta$ to predict the masked inputs $x_s$ of sample $x$. Let $f^{(i)}$ and $g^{(i)}$ denote the $i$-th MSA block of the main model $f$ and the surrogate branch $g$, respectively. Assume there are $N$ MSA blocks in total. Let $z_1^{\text{main}}$ denote the output for masked input $x_s$ of the first MSA layer $f^{(1)}$ of $f$, *i.e.*, $z_1^{\text{main}} = f^{(1)}(x_s)$. The forward process of the frozen main model $f$ with the side-tuning branch $g$ is:

$$z_i^{\text{main}} = f^{(i)}(z_{i-1}^{\text{main}}), \quad z_i^{\text{surr}} = g^{(i)}(\text{FC}^{(i)}(z_i^{\text{main}})). \tag{5}$$

After $N$ MSA blocks, an FC head is applied to generate the prediction for the partial information, *i.e.*, $y^{\text{surr}} = \text{FC}_{\text{head}}(z_N^{\text{surr}})$. The convergence of the surrogate model is analyzed as follows:

**Theorem 1** (Proof in Appendix B)**.** *Let the surrogate model be trained using gradient descent with step size $\alpha$ for $t$ iterations. The expected KL divergence between the original model's predictions $f(x)$ and the surrogate model's predictions $g_\beta(x_s)$ is upper-bounded by:*

$$\mathbb{E}_{x\sim p(x), s\sim \mathcal{U}(0,d)}\left[D_{KL}\left(f(x)\|g_\beta(x_s)\right)\right] \leq \frac{1}{2\mu}(1-\mu\alpha)^t\left(\mathcal{L}_{surr}(\beta_0) - \mathcal{L}_{surr}^\star\right), \tag{6}$$

*where $\beta_0$ is the initial parameter value, $\mathcal{L}_{surr}^\star$ is the optimal value during optimization, and $\mu$ is the minimal eigenvalue of the Hessian of $\mathcal{L}_{surr}$.*

Theorem 1 establishes a theoretical guarantee that a side-tuned surrogate can achieve performance comparable to that of a fully trained surrogate. The detailed proof is provided in Appendix B.

Figure 3(b) shows the pipeline of obtaining the surrogate in stage 1. An intuitive performance comparison of prediction models is presented in Figure 4(a). *AutoGnothi*'s surrogate surpasses the

Table 2: Inference efficiency comparison of various models. We assessed the computational cost (FLOPs), total parameters, and inference time for different methods. For the baseline method (Covert et al., 2022), we calculated these values separately for the classifier and explainer, and then combined them. In contrast, for *AutoGnothi*, we computed these values directly from our self-interpretable models, who can generate both predictions and explanations simultaneously.

| Dataset | | ImageNette | | | | MURA | Pet | Yelp |
|---|---|---|---|---|---|---|---|---|
| Model | | ViT-T | ViT-S | ViT-B | ViT-L | ViT-B | ViT-B | BERT-B |
| Classifier + | FLOPs (G) | 4.78 | 18.86 | 74.90 | 251.96 | 74.88 | 74.93 | 213.51 |
| Explainer | Time (ms) | 19.7 | 39.0 | 94.9 | 310.3 | 100.3 | 100.2 | 166.90 |
| (Covert et al.) | #Params (M) | 12.25 | 48.08 | 190.53 | 640.23 | 190.49 | 190.63 | 237.27 |
| Self-Interpretable | FLOPs (G) | 2.22 (-54%) | 8.73 (-54%) | 34.67 (-54%) | 122.60 (-51%) | 36.81 (-51%) | 34.67 (-54%) | 116.66 (-45%) |
| Model | Time (ms) | 15.4 (-22%) | 23.0 (-41%) | 52.9 (-44%) | 179.1 (-42%) | 56.7 (-43%) | 57.0 (-43%) | 118.55 (-29%) |
| (*AutoGnothi*) | #Params (M) | 5.68 (-54%) | 22.28 (-54%) | 88.22 (-54%) | 311.55 (-51%) | 93.65 (-51%) | 88.25 (-54%) | 126.63 (-47%) |

original classifier in terms of prediction accuracy and matches the performance of (Covert et al., 2022) when handling partial information, but with only $3\%$ trainable parameters. Additionally, *AutoGnothi*'s surrogate exhibits more robust predictions as the number of masked inputs increases. A detailed comparison of the training and memory costs on different models is provided in Table 1.

### 4.1.2 OBTAINING THE EXPLAINER

For the explainer model, *AutoGnothi* uses a similar LST feature backbone as in the surrogate model $g$, consisting of $N$ MSA blocks for feature extraction. Let $\phi^{(i)}$ represent the $i$-th MSA block of the explainer branch $\phi$. In addition to the lightweight backbone blocks in the side branch, we add $M$ extra FC layers as the explanation head. Together, the side network $\phi$ generates explanations based on the backbone features from the main branch. Let $z_1^{\text{main}}$ denote the output for input $x$ from the first MSA layer $f^{(i)}$ of the main branch $f$, *i.e.*, $z_1^{\text{main}} = f^{(1)}(x)$. The forward process of the explainer is:

$$z_i^{\text{main}} = f^{(i)}(z_{i-1}^{\text{main}}), \quad z_i^{\text{exp}} = \phi^{(i)}(\text{FC}^{(i)}(z_i^{\text{main}})) \quad \forall i \in \{1, \ldots, N\},$$
$$\phi^{\text{exp}}(x) = \text{FC}_{\text{head}}^{(M)}\left(\text{FC}_{\text{head}}^{(M-1)}\left(\ldots\left(\text{FC}_{\text{head}}^{(1)}\left(z_N^{\text{exp}}\right)\right)\right)\right), \tag{7}$$

where the main branch $f$ remains uncontaminated. We provide a theoretical guarantee for the convergence of the trained side branch $\phi$ as follows:

**Theorem 2** (Proof in Appendix B). *Let $\phi_v(x|y)$ denote the exact Shapley value for input-output pair $(x, y)$ in game $v$. The expected regression loss $\mathcal{L}_{exp}(\theta)$ upper bounds the Shapley value estimation error as follows,*

$$\mathbb{E}_{p(x,y)}\left[\left|\left|\phi_\theta(x, y) - \phi_v(x|y))\right|\right|_2\right] \leq \sqrt{2H_{d-1}\left(\mathcal{L}_{exp}(\theta) - \mathcal{L}_{exp}^\star\right)}, \tag{8}$$

*where $\mathcal{L}_{exp}^\star$ represents the optimal loss achieved by the exact Shapley values, and $H_{d-1}$ is the $(d-1)$-th harmonic number.*

Theorem 2 provides a theoretical guarantee that a side-tuned explainer can achieve performance on par with a fully trained explainer. The complete proof is presented in Appendix B.

Figure 3(b) shows the pipeline of obtaining the explainer in stage 2. We evaluated the explanation quality of *AutoGnothi* against various baselines, as shown in Figure 4(b). *AutoGnothi* achieved the highest insertion and lowest deletion scores among 12 explanation methods, demonstrating superior explanation quality. Compared to (Covert et al., 2022), we reduced the trainable parameters for explainers by $98\%$ while maintaining comparable or even superior interpretability. Table 1 provides detailed comparisons of training and memory efficiency for different models.

### 4.1.3 GENERATING EXPLANATION FOR BLACK-BOX MODELS

After obtaining the explainer, we now detail the explanation procedure. In most post-hoc explanation methods (Selvaraju et al., 2020; Chattopadhay et al., 2018; Binder et al., 2016; Covert et al., 2022), predictions and explanations must be computed separately for a single input. For instance, as illustrated in Figure 3(a), two separate inferences are required to explain a single prediction. In contrast, as shown in Figure 3(b), *AutoGnothi* generates both predictions and explanations simultaneously, needing only one inference. To evaluate this efficiency, we measured inference time, computational cost (FLOPs), and total parameters required to generate predictions and explanations for different methods. The comparison of inference efficiency is highlighted in Table 2.

## 4.2 Difference between *AutoGnothi* and previous PETL Methods

In this section, we provide an empirical analysis of why *AutoGnothi* outperforms previous PETL approaches. We highlight that *AutoGnothi* is not a simple extension of classical PETL, where full fine-tuning is typically the optimal baseline. Additionally, *AutoGnothi* exploits the intrinsic correlation between the prediction task and its explanation, enabling black-box models to become self-interpretable without sacrificing prediction accuracy. We elaborate on these two points below.

**Full fine-tuning poses challenges for achieving self-interpretability and is not the optimal baseline for *AutoGnothi*.** For classical PETL, the goal is often to match the performance of fully fine-tuned models on standard tasks (Houlsby et al., 2019; Chen et al., 2023b; Hu et al., 2021; Mercea et al., 2024). However, in XAI scenarios, the challenge shifts: it becomes difficult, if not impossible, to train a model that balances both prediction accuracy and explanation quality (Arrieta et al., 2020; Gunning et al., 2019; Došilović et al., 2018). In fact, fine-tuning models to adapt interpretability without forgetting pre-trained knowledge can be difficult (Li & Hoiem, 2017). Additionally, full fine-tuning the original model also puts us in the *Theseus's Paradox*: we won't be sure if we are explaining the very same model anymore. Even if full fine-tuning were practical, it would contradict the goal of interpreting the pre-trained model. In contrast, *AutoGnothi* pipeline addresses this issue by freezing the primary model and training only a side network to generate explanations, offering an efficient solution that enables self-interpretability without degrading prediction performance.

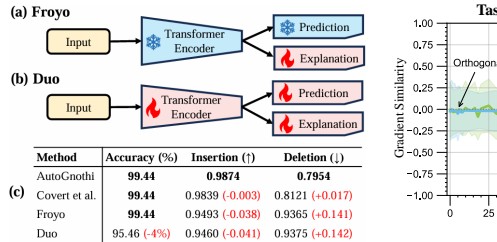

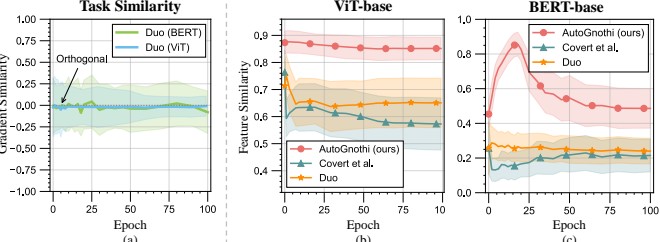

Figure 5: Other pipelines to achieve the self-interpretability through the full fine-tuning. (a) Freeze the transformer encoder and prediction head, learning only the explanation head. (b) Simultaneously learn the transformer encoder, and both task heads. (c) Comparison of classification and explanation performance between different pipelines for ViT-base.

Figure 6: Explanation of why the Duo pipeline underperforms compared to *AutoGnothi*. (a) The full fine-tuning strategy employed by Duo is not the optimal baseline for *AutoGnothi* due to significant gradient conflicts between the prediction and explanation tasks. This conflict results in degraded performance, raising concerns about whether the explanation truly pertains to the same model. Note that gradient similarity can only be measured for the Duo pipeline, as other methods freeze the prediction backbones. (b) In contrast, *AutoGnothi* exhibits a stronger correlation between features of the prediction and explanation tasks. We measured the feature similarity with CKA.

Building on prior work that highlights the challenges of achieving self-interpretability through full fine-tuning, we conducted experiments to further explore this issue. As depicted in Figure 5(a)(b), we introduce two additional pipelines, *Froyo* and *Duo*. *Froyo* adds an explanation head while keeping the transformer encoder and prediction head frozen to preserve prediction accuracy. In contrast, *Duo* jointly learns both the prediction task and its explanation. Both pipelines use the same encoder, prediction head, and explanation head architectures as described in (Covert et al., 2022).

We performed experiments using ViT-base model trained on the ImageNette dataset. Our findings show that the *Froyo* pipeline underperforms due to the limited trainable parameters in the explanation head, resulting in degraded explanation quality. As illustrated in Figure 5(c), this led to a reduction in insertion by $0.038$ and an increase in deletion by $0.141$, despite no impact on prediction accuracy.

For the Duo pipeline, we observed a $4.0\%$ decline in prediction accuracy on the ViT-base model, accompanied by a reduction in insertion by $0.041$ and an increase in deletion by $0.142$. Further empirical evidence, as depicted in Figure 6(a), highlights conflicting gradients between the prediction and explanation tasks during training. Additionally, as previously discussed, the *Theseus's Paradox* arises when changes in predictions result in evolving explanations, thereby challenging the consistency and identity of the original model.

***AutoGnothi* uncovers the intrinsic correlation between predictions and explanations.** While the *AutoGnothi* pipeline enables self-interpretation with superior efficiency, the underlying mechanisms connecting prediction and explanation remain underexplored. We propose that *AutoGnothi* leverages the intrinsic relationship between the backbone features used for both tasks. This correlation is illustrated in Figure 6(b), where we evaluated the Central Kernel Alignment (CKA) (Kornblith et al., 2019) between the backbone features of the original pre-trained model and those of various explainers. Our results show that *AutoGnothi* exhibits higher feature similarity between prediction and explanation tasks on ViT-base and BERT-base models, supporting our hypothesis.

## 5 EXPERIMENTS

### 5.1 EXPERIMENTAL SETTINGS

**Datasets and Black-box Models.** For image classification, we used the ImageNette (Howard & Gugger, 2020), Oxford IIIT-Pets (Parkhi et al., 2012), and MURA (Rajpurkar et al., 2017) datasets, following (Covert et al., 2022). For sentiment analysis, we utilized the Yelp Review Polarity dataset (Zhang et al., 2015). In terms of black-box models, we employed the widely used ViT models (Dosovitskiy, 2020) for vision tasks, including ViT-tiny, ViT-small, ViT-base, and ViT-large. For language tasks, we used the BERT-base model (Devlin et al., 2019).

**Implementation Details.** For surrogates and explainers, *AutoGnothi* incorporates the same number of MSA blocks as the black-box model being explained in the side network and utilizes a reduction factor of $r = 8$ for the lightweight side branch on both the ImageNette and Oxford IIIT-Pets datasets, and $r = 4$ for MURA and Yelp Review Polarity. Surrogates use the same task head as the black-box classifiers with one additional FC layer as classification head for handling partial information. Explainers utilize three additional FC layers as the explanation head after the side network backbone. For attention masking, we employed a causal attention masking strategy, setting attention values to a large negative number before applying the softmax operation (Brown, 2020). More detailed training settings are provided in Appendix A.

**Evaluation Metrics for Explanations.** We used the widely adopted insertion and deletion metrics (Petsiuk, 2018) to evaluate explanation quality. These metrics are computed by progressively

Table 3: Quality metrics (insertion and deletion) for target class explanations of ViT-base across baseline methods and *AutoGnothi* on the ImageNette dataset. More results for other datasets and models are provided in Appendix C.

| Method | Insertion (↑) | Deletion (↓) |
|---|---|---|
| Random | $0.9578_{\pm 0.0790}$ | $0.9584_{\pm 0.0764}$ |
| Attention last | $0.9633_{\pm 0.0659}$ | $0.8524_{\pm 0.1748}$ |
| Attention rollout | $0.9408_{\pm 0.0834}$ | $0.9168_{\pm 0.1277}$ |
| GradCAM (Attn) | $0.9447_{\pm 0.0936}$ | $0.9562_{\pm 0.0916}$ |
| GradCAM (LN) | $0.9343_{\pm 0.0829}$ | $0.9426_{\pm 0.1307}$ |
| Vanilla (Pixel) | $0.9487_{\pm 0.0688}$ | $0.8945_{\pm 0.1513}$ |
| Vanilla (Embed) | $0.9563_{\pm 0.0643}$ | $0.8618_{\pm 0.1754}$ |
| IntGrad (Pixel) | $0.9670_{\pm 0.0575}$ | $0.9408_{\pm 0.1141}$ |
| IntGrad (Embed) | $0.9670_{\pm 0.0575}$ | $0.9408_{\pm 0.1141}$ |
| SmoothGrad (Pixel) | $0.9591_{\pm 0.0760}$ | $0.8459_{\pm 0.1788}$ |
| SmoothGrad (Embed) | $0.9529_{\pm 0.0931}$ | $0.9561_{\pm 0.0764}$ |
| VarGrad (Pixel) | $0.9616_{\pm 0.0725}$ | $0.8600_{\pm 0.1692}$ |
| VarGrad (Embed) | $0.9552_{\pm 0.0901}$ | $0.9568_{\pm 0.0756}$ |
| LRP | $0.9677_{\pm 0.0623}$ | $0.8393_{\pm 0.1866}$ |
| Leave-one-out | $0.9696_{\pm 0.0353}$ | $0.9334_{\pm 0.1493}$ |
| RISE | $0.9772_{\pm 0.0225}$ | $0.8959_{\pm 0.1962}$ |
| Covert et al. | $0.9839_{\pm 0.0375}$ | $0.8121_{\pm 0.1768}$ |
| *AutoGnothi* (Ours) | $\mathbf{0.9874_{\pm 0.0265}}$ | $\mathbf{0.7954_{\pm 0.2294}}$ |

inserting or deleting features based on their importance and observing the impact on the model's predictions. The corresponding surrogate model trained to handle partial inputs is used for this process to generate prediction for masked inputs. We calculated the area under the curve (AUC) for the predictions and average results for randomly selected $1,000$ samples on all datasets.

**Baseline Methods.** We considered 12 representative explanation methods for comparison. For attention-based methods, we utilized attention rollout and attention last (Abnar & Zuidema, 2020). For gradient-based methods, we used Vanilla Gradients (Simonyan, 2013), IntGrad (Sundararajan et al., 2017), SmoothGrad (Smilkov et al., 2017), VarGrad (Hooker et al., 2019), LRP (Binder et al., 2016), and GradCAM (Selvaraju et al., 2020). For removal-based methods, we employed leave-one-out (Zeiler & Fergus, 2014) and RISE (Petsiuk, 2018). For Shapley value-based methods, we utilized KernelSHAP (Lundberg, 2017) and ViT Shapley (Covert et al., 2022) as baselines.

### 5.2 EVALUATING TRAINING, MEMORY AND INFERENCE EFFICIENCY

To evaluate the training and memory efficiency of *AutoGnothi*, we compared it with various baselines in terms of trainable parameters and memory usage. Table 1 provides a summary of the memory

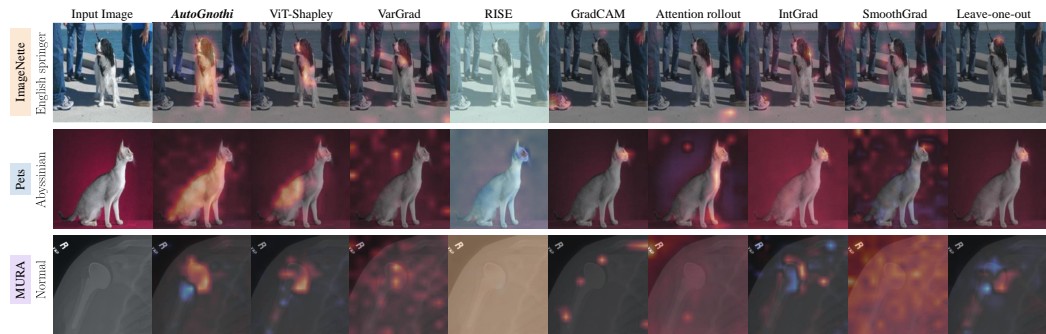

Figure 7: Visualization of ViT explanations on the ImageNette, Oxford-IIIT Pets, and MURA datasets. *AutoGnothi* qualitatively outperforms other baseline approaches.

costs and trainable parameters for both surrogate and explainer models across different methods. During training, *AutoGnothi* achieves a significant reduction of over $87\%$ in trainable parameters and more than $65\%$ in GPU memory usage for both surrogates and explainers on both vision and language datasets, while maintaining competitive performance in both prediction accuracy and explanation quality. The memory cost is evaluated with training batch size $= 1$.

Next, we evaluate the inference efficiency of our self-interpretable models by comparing the computational cost (FLOPs), inference time, and the total number of parameters required for generating both predictions and explanations. As shown in Table 2, *AutoGnothi* significantly reduces inference time and FLOPs compared to baseline models that require separate inferences for predictions and explanations. Specifically, when both the predictions and the explanations are required, *AutoGnothi* is capable of reducing at least $45\%$ in FLOPs, $22\%$ in inference time (up to $44\%$), and $47\%$ in total parameters end-to-end across all datasets and tasks.

### 5.3    EVALUATING EXPLANATION QUALITY

**Quantitative Results.** To evaluate the quality of explanations generated by *AutoGnothi*, we compared it against 12 state-of-the-art baseline methods using the insertion and deletion metrics. Table 3 presents results from a ViT-base model trained on the ImageNette dataset, where *AutoGnothi* consistently achieves the best insertion and deletion scores for target class explanations across various datasets. Further detailed results for other models and tasks are provided in Appendix C. For vision task, please refer to Tables 4, 5, and 6 for the results of ViT-tiny, ViT-small, and ViT-large on ImageNette, respectively. Results for ViT-base on Oxford-IIIT Pets are provided in Table 7, and results for ViT-base on Mura are shown in Table 8. For language task, we also provide results of BERT-base on Yelp Reivew Polarity in Table 9.

**Qualitative Results.** We also provide visualization results for ViTs on different datasets, as shown in Figure 7. It may be observed that RISE happened to fail to provide human-interpretable or intuitive results, which amongst all others *AutoGnothi* offers more accurate explanations with a clearer focus on the subject and diluted colours for irrelevant classes. Additional visualization results for ViT-base on more datasets are provided in Appendix E, shown in Figures 8, 9, 10, 11, 12, and 13.

## 6    CONCLUSION

This paper introduces *AutoGnothi* to bridge the gap between self-interpretable models and post-hoc explanation methods in Explainable AI. Inspired by parameter-efficient transfer-learning, *AutoGnothi* incorporates a lightweight side network that allows black-box models to generate faithful Shapley value explanations without affecting the original predictions. Notably, *AutoGnothi* outperforms directly fine-tuning the all the parameters in the model for explanation by a clear margin. This approach empowers black-box models with self-interpretability, which is superior to standard post-hoc explanations that require generating predictions and explanations in two separate, heavy inferences. Experiments on ViT and BERT demonstrate that *AutoGnothi* achieves superior efficiency on computation, storage and memory in both training and inference periods.

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

# A  FURTHER EXPERIMENTAL DETAILS

## A.1  ENVIRONMENT

Our experiments were conducted on one 128-core AMD EPYC 9754 CPU with one NVIDIA GeForce RTX 4090 GPU with 24 GB VRAM. No multi-card training or inference was involved. We implemented the training and inference pipelines for image classification tasks under the PyTorch Lightning framework, and for the sentiment analysis task with just PyTorch. For evaluations on baseline methods we leverage the SHAP library (Lundberg, 2017), with minor modifications applied to bridge data format differences between Numpy and PyTorch.

## A.2  FLOPs AND MEMORY

We used the `profile` function from the `thop` library to evaluate the FLOPs for each model during the inference stage. Memory consumption was manually calculated during the training stage based on the model parameters, activations, and intermediate results. Our memory estimations were made under the assumption that the memory was evaluated under a batch size of 1 and uses 32-bit floating point precision (torch.float32).

## A.3  CLASSIFIER

The training of all tasks is split into 3 stages. In the first stage parameters of the classifier are inherited from the original base model verbatim, with the exception of *AutoGnothi* adding additional parameters for the new side branch, initialized with Kaiming initialization.

We fine-tune this classifier model on the exact same dataset with the AdamW optimizer, using a learning rate of $10^{-5}$ for 25 epochs, and retain the best checkpoint rated by minimal validation loss. Classification loss is minimum square error with respective to the predicted classes and the ground-truth labels. For *AutoGnothi* the classes come from the side branch, while all remaining models use the classes from the main branch. We train and evaluate these methods on 1 Nvidia RTX 4090, and use a batch size of 32 samples provided that it fits inside the available GPU memory.

We freeze parameters in all stages likewise. As is described in 4.2, *AutoGnothi* only trains the side branch and all parameters from the original model are frozen. In ViT-shapley (Covert et al.) and Duo pipelines, all parameters are trained, whilst the Froyo pipeline only trains the classification head.

## A.4  SURROGATE

Surrogate models have the same model architecture as the classifiers, with a different recipe. We load all parameters from the classifier without any changes or additions, further fine-tune the model for 50 epochs, and retain the best checkpoint with minimal validation loss.

Unlike the classifier, the surrogate model focuses on mimicking the classifier model's behavior under a masked context. Consider the logits $p(y|x_{\mathbf{0}})$ from the classifier model, where $x$ is the input and $y$ is the corresponding class label. The surrogate model aims at closing in its masked logits distribution, $p(y|x_s)$, with the original distribution:

$$\mathcal{L}_{\text{surr}}(\beta) = \mathop{\mathbb{E}}_{x \sim p(x), s \sim \mathcal{U}(0,d)} \left[ D_{\text{KL}} \left( f(x) \| g_\beta(x_s) \right) \right], \tag{1}$$

The mask is selected on an equi-categorical basis. We first pick an integer $n_s = s \cdot \mathbf{1}^\top$ at uniform distribution, denoting the number of tokens that shall be masked. $n_s$ mutually exclusive indices are then randomly chosen from the input at uniform distribution. To avoid inhibiting model capabilities, special tokens like the implicit class token in ViT or the `[CLS]` token applied by the BERT tokenizer are never masked.

In order to selectively hide or mask inputs from the model, we apply causal attention masks for both the image models and text models in our experiments. However, it's worth noting that while they may confuse the transformer's attention mechanisms, certain other methods are also capable of concealing these input tokens, primarily zeroing or assigning random values to the said pixels

in image models, or assigning `[PAD]` and `[MASK]` to the selected tokens. We follow prior work (Covert et al., 2022) and adhere to causal attention masks.

Recall that the transformer self-attention mechanism used in ViT (Dosovitskiy, 2020), BERT (Devlin et al., 2019). Given an attention input $t \in \mathbb{R}^{d \times h}$ and self-attention parameters $W_{\text{qkv}} \in \mathbb{R}^{h \times 3h'}$, whereas $d$ is the number of tokens and $h$ is the attention's hidden size, and $h'$ be the size of each attention head, the output of the self-attention $\text{SA}(t)$ for a single head is computed as follows:

$$[Q, K, V] = u \cdot W_{\text{qkv}} \tag{2}$$

$$A = \text{softmax}\left(\frac{Q \cdot K^\top}{\sqrt{h'}}\right) \tag{3}$$

$$\text{SA}(t) = A \cdot V \tag{4}$$

Transformers in practice use a multiple $k$ attention heads, holding that $k \cdot h' = h$. An attention projection matrix $P_{\text{msa}} \in \mathbb{R}^{h \times h}$ is used to combine the outputs of all attention heads. Denoting the $i$-th self-attention head's output as $SA_i(t)$, the final output of the attention layer $\text{MSA}(t)$ is thus computed:

$$\text{MSA}(t) = [\text{SA}_1(t), \text{SA}_2(t), \ldots, \text{SA}_n(t)] \cdot P_{\text{msa}} \tag{5}$$

We notice that the attention mechanism is entirely unrelated to the number of tokens, and can operate in the absence of certain input tokens. Let an indicator vector $s \in \{0, 1\}^d$ correspond to a subset of the input tokens (applying equally to images and text tokens), we calculate the masked self-attention over $t$ and $s$ as follows:

$$[Q, K, V] = u \cdot W_{Q,K,V} \tag{6}$$

$$A = \text{softmax}\left(\frac{Q \cdot K^\top - (1 - s) \cdot \infty}{\sqrt{h'}}\right) \tag{7}$$

$$\text{SA}(t, s) = A \cdot V \tag{8}$$

We apply masking to the multi-head attention layers likewise, such that:

$$\text{MSA}(t, s) = [\text{SA}_1(t, s), \text{SA}_2(t, s), \ldots, \text{SA}_n(t, s)] \cdot P_{\text{msa}} \tag{9}$$

This mechanism is widely implemented for attention models in commonplace libraries such as HuggingFace's Transformers, named by the argument `attention_mask` in the input tensor.

## A.5 EXPLAINER

We load explainer model parameters from surrogate model checkpoints, such that all are copied from the surrogate model to the explainer model, except for the last classification head, which is replaced with an explainer head. The explainer head contains 3 MLP layers and a final linear layer, with GeLU (Hendrycks & Gimpel, 2017) activations from in between. For *AutoGnothi* only the classification head on the side branch is replaced. We train the explainer model for 100 epochs with the AdamW optimizer, using a learning rate of $10^{-5}$, and keep the best checkpoint.

In our implementation, we took 2 input images in each mini-batch and generated 16 random masks for each image, resulting in a parallelism of 32 instances per batch. A slight change is applied to the masking algorithm in the explainer model from the surrogate model, in order to reduce variance during gradient descent. Specifically, in addition to generating masks uniform, we follow prior work (Covert et al., 2022) and use the paired sampling trick (Covert & Lee, 2020), pairing each subset $s$ with its complement $\mathbf{1} - s$. This algorithm equally applies to both image and text classification models.

The explainer model is trained to approximate the Shapley value $\phi_\theta(x, y)$. Let $g_\beta(x_s | y)$ be the surrogate values respective to the input $x$ and the class $y$, masked by the indicator vector $s$, and

$g_\beta(x_\mathbf{0}|y)$ be the surrogate values without masking. Following (Covert et al., 2022), we minimize the following loss function:

$$\mathcal{L}_{exp}(\theta) = \mathop{\mathbb{E}}_{(x,y)\sim p(x,y), s\sim q(s)} \left[ \left( g_\beta(x_s|y) - g_\beta(x_\mathbf{0}|y) - s^\top \phi_\theta(x,y) \right)^2 \right] \tag{10}$$

$$\text{s.t.} \quad \mathbf{1}^\top \phi_\theta(x,y) = g_\beta(x_\mathbf{1}|y) - g_\beta(x_\mathbf{0}|y) \quad \forall(x,y), \tag{Efficiency}$$

Notice that the explainer model $\phi_\theta(x,y)$ is being trained under a mean squared error loss, and that 16 random masks are generated for each image in the mini-batch. The explainer is hence optimized against a distribution of the Shapley values, so an accurate calculation of ground-truth Shapley values for each training sample is not even remotely necessary.

Also, the aforementioned efficiency constraint is necessary for the explainer model to output faithful and exact Shapley values. We leverage *additive efficient normalization* from (Ruiz et al., 1998) and use the same approach as prior work (Jethani et al., 2021; Covert et al., 2022) to enforce this constraint. The model is trained to make unconstrained predictions as is described in equation 10, which we then modify using the following transformation to have it constrained:

$$\phi_\theta(x,y) \leftarrow \phi_\theta(x,y) + \frac{g_\beta(x_\mathbf{1}|y) - g_\beta(x_\mathbf{0}|y) - \mathbf{1}^T \cdot \phi_\theta(x,y)}{d} \tag{11}$$

After training the explainer model, we merge the original classifier model and all relevant intermediate stages' models into one single, independent model to contain both the classification task and the explanation task to have them run concurrently. For the baseline method, ViT-shapley and its modified counterpart for NLP, no parameters overlap between the two tasks, thus inference must be done on both tasks resulting in a huge implied performance overhead. *AutoGnothi*, however, only requires a validation pass to ensure that the original classifier head is preserved verbatim in the final model. This is done by comparing the output of the original classifier and the final model on any arbitrary input.

## A.6 DATASETS

In this section we explain in more detail which datasets are selected and how they are processed for our experiments. Three datasets are used for the image classification task. The ImageNette dataset includes $9,469$ training samples and $3,925$ validation samples for 10 classes. MURA (musculoskeletal radiographs) has $36,808$ training samples and $3,197$ validation samples for 2 classes. The Oxford-IIIT Pets dataset contains $5,879$ training samples, $735$ validation samples and $735$ test samples in 37 classes. For the text classification (sequence classification) task, we use the Yelp Polarity dataset, which originally contains $560,000$ training samples and $38,000$ test samples.

For each epoch, image classifiers (ViT) iterate through all available images in either of the train or test dataset. Specifically to during training, images are normalized by the mean value and standard deviation of each corresponding training dataset, before being down-sampled to $224 \times 224$ pixels. For text classifiers, each of the epoch is trained on exactly 2048 training samples randomly chosen from the dataset, and validated on 256 equally random test samples. This is primarily done to serve our needs in frequent checkpoints for more thorough data analysis such as on CKA or parameter gradients.

Due to the sheer cost from some metrics on certain tasks, we reduced the size of our test set during evaluation. We selected 1000 test samples for datasets ImageNette, MURA and Yelp Review Polarity, and randomly selected 300 samples for the Oxford-IIIT Pets dataset. For each dataset this subset stays the same between different explanation methods and different model sizes. We emphasize that these samples are deliberately independent from the training set to avoid potential bias from the results.

# B   PROOFS OF THEOREMS

Here we provide detailed proof for Theorem 1 and Theorem 2, which provide theoretical guarantee for the performance of surrogates and explainers. Our proof follows (Simon & Vincent, 2020) and (Covert et al., 2022), which exhibit similar results for a single data point.

**Lemma 1.** *For a single input $x$, the expected loss under Eq. (2) is $\mu$-strongly convex, where $\mu$ is the minimal eigenvalue of the Hessian of $\mathcal{L}_{surr}(\beta)$.*

*Proof.* The expected loss for a single input $x$ under the new objective function is given by:

$$\mathcal{L}_{\text{surr}}(\beta) = \mathbb{E}_{s \sim \mathcal{U}(0,d)} \left[ D_{\text{KL}}(f(x) \| g_\beta(x_s)) \right]. \tag{12}$$

This loss function is convex in $\beta$ because the KL divergence $D_{\text{KL}}(f(x)\|g_\beta(x_s))$ is convex in $g_\beta(x_s)$, and $g_\beta(x_s)$ is a smooth function of $\beta$. The Hessian of $\mathcal{L}_{\text{surr}}(\beta)$ with respect to $\beta$ is:

$$\nabla_\beta^2 \mathcal{L}_{\text{surr}}(\beta) = \mathbb{E}_{s \sim \mathcal{U}(0,d)} \left[ \nabla_\beta^2 D_{\text{KL}}(f(x)\|g_\beta(x_s)) \right]. \tag{13}$$

The convexity of $\mathcal{L}_{\text{surr}}(\beta)$ is determined by the smallest eigenvalue of this Hessian, $\mu$. Since the KL divergence is strictly convex, the minimum eigenvalue is positive, which implies $\mu$-strong convexity. $\square$

**Theorem 1.** *Let the surrogate model be trained using gradient descent with step size $\alpha$ for $t$ iterations. The expected KL divergence between the original model's predictions $f(x)$ and the surrogate model's predictions $g_\beta(x_s)$ is upper-bounded by:*

$$\mathbb{E}_{x \sim p(x), s \sim \mathcal{U}(0,d)} \left[ D_{KL}(f(x)\|g_\beta(x_s)) \right] \leq \frac{1}{2\mu} (1 - \mu\alpha)^t \left( \mathcal{L}_{surr}(\beta_0) - \mathcal{L}_{surr}^\star \right), \tag{14}$$

*where $\beta_0$ is the initial parameter value, and $\mathcal{L}_{surr}^\star$ is the optimal value during optimization.*

*Proof.* Let the surrogate model be trained using gradient descent with step size $\alpha$ for $t$ iterations. The optimization process for minimizing the expected KL divergence between $f(x)$ and $g_\beta(x_s)$ can be written as:

$$\beta_{t+1} = \beta_t - \alpha \nabla_\beta \mathcal{L}_{\text{surr}}(\beta_t). \tag{15}$$

Because $\mathcal{L}_{\text{surr}}(\beta)$ is $\mu$-strongly convex, we can apply the standard result for gradient descent convergence on strongly convex functions, which gives the following bound:

$$\mathcal{L}_{\text{surr}}(\beta_t) - \mathcal{L}_{\text{surr}}^\star \leq (1 - \mu\alpha)^t \left( \mathcal{L}_{\text{surr}}(\beta_0) - \mathcal{L}_{\text{surr}}^\star \right), \tag{16}$$

where $\mathcal{L}_{\text{surr}}^\star$ is the optimal value, and $\beta_0$ is the initial parameter value.

Since the KL divergence is bounded by the expected loss, we have:

$$\mathbb{E}_{x \sim p(x), s \sim \mathcal{U}(0,d)} \left[ D_{\text{KL}}(f(x)\|g_\beta(x_s)) \right] \leq \mathcal{L}_{\text{surr}}(\beta_t). \tag{17}$$

Substituting the bound on $\mathcal{L}_{\text{surr}}(\beta_t)$, we obtain:

$$\mathbb{E}_{x \sim p(x), s \sim \mathcal{U}(0,d)} \left[ D_{\text{KL}}(f(x)\|g_\beta(x_s)) \right] \leq \frac{1}{2\mu} (1 - \mu\alpha)^t \left( \mathcal{L}_{\text{surr}}(\beta_0) - \mathcal{L}_{\text{surr}}^\star \right). \tag{18}$$

$\square$

**Lemma 2.** *For a single input-output pair $(x, y)$, the expected loss under Eq.(3) for the prediction $\phi_\theta(x, y)$ is $\mu$-strongly convex with $\mu = H_{d-1}^{-1}$, where $H_{d-1}$ is the $(d-1)$-th harmonic number.*

*Proof.* For an input-output pair $(x, y)$, the expected loss for the prediction $\phi = \phi_\theta(x, y)$ is defined as

$$
\begin{aligned}
L_\theta(x, y) &= \mathbb{E}_{s \sim p(s)} \left[ \left( g_\beta(x_s|y) - g_\beta(x_0|y) - s^\top \phi \right)^2 \right] \\
&= \phi^\top \mathbb{E}_{s \sim p(s)}[ss^\top]\phi - 2\mathbb{E}_{s \sim p(s)} \left[ s \left( g_\beta(x_s|y) - g_\beta(x_0|y) \right) \right] \phi \\
&\quad + \mathbb{E}_{s \sim p(s)} \left[ \left( g_\beta(x_s|y) - g_\beta(x_0|y) \right)^2 \right].
\end{aligned}
\tag{19}
$$

This is a quadratic function of $\phi$ with its Hessian given by

$$\nabla_\theta^2 L_\theta(x, y) = 2 \cdot \mathbb{E}_{s \sim p(s)}[ss^\top]. \tag{20}$$

The eigenvalues of the Hessian determine the convexity of $L_\theta(x, y)$, and the entries of the Hessian can be derived from the subset distribution $p(s)$. The distribution assigns equal probability to subsets with the same cardinality, thus we define the shorthand $p_k \equiv p(s)$ for $s$ such that $\mathbf{1}^\top s = k$. Specifically, we have:

$$p_k = Q^{-1} \frac{d-1}{\binom{d}{k} k(d-k)} \quad \text{and} \quad Q = \sum_{k=1}^{d-1} \frac{d-1}{k(d-k)}. \tag{21}$$

We can then write $A \equiv \mathbb{E}_{s \sim p(s)}[ss^\top]$ and derive its entries as follows:

$$A_{ii} = \Pr(s_i = 1) = \sum_{k=1}^{d} \binom{d-1}{k-1} p_k$$
$$= Q^{-1} \sum_{k=1}^{d-1} \frac{d-1}{d(d-k)} = \frac{\sum_{k=1}^{d-1} \frac{d-1}{d(d-k)}}{\sum_{k=1}^{d-1} \frac{d-1}{k(d-k)}} \tag{22}$$

$$A_{ij} = \Pr(s_i = s_j = 1) = \sum_{k=2}^{d} \binom{d-2}{k-2} p_k$$
$$= Q^{-1} \sum_{k=2}^{d-1} \frac{k-1}{d(d-k)} = \frac{\sum_{k=2}^{d-1} \frac{k-1}{d(d-k)}}{\sum_{k=1}^{d-1} \frac{d-1}{k(d-k)}}. \tag{23}$$

Based on this, we observe that $A$ has the structure $A = (b-c)I_d + c\mathbf{1}\mathbf{1}^\top$, where $b \equiv A_{ii} - A_{ij}$ and $c \equiv A_{ij}$. Following (Simon & Vincent, 2020; Covert et al., 2022), the minimum eigenvalue is given by $\lambda_{\min}(A) = b - c$. A more detailed derivation reveals that it depends on the $(d-1)$-th harmonic number, $H_{d-1}$:

$$\lambda_{\min}(A) = b - c = A_{ii} - A_{ij}$$
$$= \frac{\sum_{k=1}^{d-1} \frac{d-1}{d(d-k)}}{\sum_{k=1}^{d-1} \frac{d-1}{k(d-k)}} - \frac{\sum_{k=2}^{d-1} \frac{k-1}{d(d-k)}}{\sum_{k=1}^{d-1} \frac{d-1}{k(d-k)}}$$
$$= \frac{\frac{1}{d} + \sum_{k=2}^{d-1} \frac{d-k}{d(d-k)}}{\sum_{k=1}^{d-1} \frac{d-1}{k(d-k)}} = \frac{\frac{1}{d} + \frac{d-2}{d}}{\sum_{k=1}^{d-1} \frac{d-1}{k(d-k)}} \tag{24}$$
$$= \frac{d-1}{d} \cdot \frac{1}{\sum_{k=1}^{d-1} \frac{d-1}{k(d-k)}} = \frac{1}{2 \sum_{k=1}^{d-1} \frac{1}{k}} = \frac{1}{2 H_{d-1}}.$$

The minimum eigenvalue is therefore strictly positive, implying that $L_\theta(x, y)$ is $\mu$-strongly convex, where $\mu$ is given by

$$\mu = 2 \cdot \lambda_{\min}(A) = H_{d-1}^{-1}. \tag{25}$$

Note that the strong convexity constant $\mu$ is independent of $(x, y)$ and is determined solely by the number of input variables $d$. □

**Theorem 2.** *Let $\phi_v(x|y)$ denote the exact Shapley value for input-output pair $(x, y)$ in game $v$. The expected regression loss $\mathcal{L}_{exp}(\theta)$ upper bounds the Shapley value estimation error as follows,*

$$\mathbb{E}_{(x,y) \sim p(x,y)} \left[ \left|\left| \phi_\theta(x, y) - \phi_v(x|y)) \right|\right|_2 \right] \leq \sqrt{2 H_{d-1} \left( \mathcal{L}_{exp}(\theta) - \mathcal{L}_{exp}^\star \right)}, \tag{26}$$

*where $\mathcal{L}_{exp}^\star$ represents the optimal loss achieved by the exact Shapley values.*

*Proof.* We begin by considering a single input-output pair $(x, y)$, where the prediction is given by $\phi = \phi_\theta(x, y; \theta)$. To account for the linear constraint (the Shapley value's efficiency constraint) in our objective, we write the Lagrangian $L_\theta(x, y, \gamma)$:

$$L_\theta(x, y, \gamma) = L_\theta(x, y) + \gamma \left( g_\beta(x_{\mathbf{1}}|y) - g_\beta(x_{\mathbf{0}}|y) - \mathbf{1}^\top \phi \right), \tag{27}$$

where $\gamma \in \mathbb{R}$ is the Lagrange multiplier. The Lagrangian $L_\theta(x, y, \gamma)$ is $\mu$-strongly convex, sharing the same Hessian as $L_\theta(x, y)$:

$$\nabla_\theta^2 L_\theta(x, y, \gamma) = \nabla_\theta^2 L_\theta(x, y). \tag{28}$$

By strong convexity, we can bound the distance between $\phi$ and the global minimizer using the Lagrangian's value. Let $(\theta^\star, \gamma^\star)$ be the optimizer of the Lagrangian, such that

$$\phi_{\theta^\star}(x, y) = \phi_v(x|y), \tag{29}$$

where $\phi_v(x|y)$ is the exact Shapley value.

From the first-order condition of strong convexity, we obtain the inequality:

$$L_\theta(x, y, \gamma^\star) \geq L_{\theta^\star}(x, y, \gamma^\star) + (\phi - \phi_{\theta^\star}(x, y))^\top \nabla_\theta L_{\theta^\star}(x, y, \gamma^\star) + \frac{\mu}{2}\|\phi - \phi_{\theta^\star}(x, y)\|_2^2. \tag{30}$$

By the KKT conditions, $\nabla_\theta L_{\theta^\star}(x, y, \gamma^\star) = 0$, so the inequality simplifies to:

$$L_\theta(x, y, \gamma^\star) \geq L_{\theta^\star}(x, y, \gamma^\star) + \frac{\mu}{2}\|\phi - \phi_{\theta^\star}(x, y)\|_2^2. \tag{31}$$

Rearranging this, we get:

$$\|\phi - \phi_{\theta^\star}(x, y)\|_2^2 \leq \frac{2}{\mu} \left( L_\theta(x, y, \gamma^\star) - L_{\theta^\star}(x, y, \gamma^\star) \right). \tag{32}$$

Since $\phi$ is a feasible solution (i.e., it satisfies the linear constraint), this further simplifies to:

$$\|\phi - \phi_{\theta^\star}(x, y)\|_2^2 \leq \frac{2}{\mu} \left( L_\theta(x, y) - L_{\theta^\star}(x, y) \right). \tag{33}$$

Next, we take the expectation over $(x, y) \sim p(x, y)$. Denote the expected regression loss as $\mathcal{L}_{exp}(\theta)$, which is:

$$\mathcal{L}_{\exp}(\theta) = \mathbb{E}_{(x,y)\sim p(x,y)} \left[ \left( g_\beta(x_s|y) - g_\beta(x_{\mathbf{0}}|y) - s^\top \phi_\theta(x, y; \theta) \right)^2 \right]. \tag{34}$$

Let $\mathcal{L}_{\exp}^\star$ denote the loss achieved by the exact Shapley values. Taking the bound from the previous inequality in expectation, we have:

$$\mathbb{E}_{(x,y)\sim p(x,y)} \left[ \|\phi_\theta(x, y; \theta) - \phi_v(x|y)\|_2^2 \right] \leq \frac{2}{\mu} \left( \mathcal{L}_{\exp}(\theta) - \mathcal{L}_{\exp}^\star \right). \tag{35}$$

Finally, applying Jensen's inequality to the left-hand side, we obtain:

$$\mathbb{E}_{(x,y)\sim p(x,y)} \left[ \|\phi_\theta(x, y; \theta) - \phi_v(x|y)\|_2 \right] \leq \sqrt{\frac{2}{\mu} \left( \mathcal{L}_{\exp}(\theta) - \mathcal{L}_{\exp}^\star \right)}. \tag{36}$$

Substituting the value of $\mu = 1/H_{d-1}$ from Lemma 2, we conclude that:

$$\mathbb{E}_{(x,y)\sim p(x,y)} \left[ \|\phi_\theta(x, y; \theta) - \phi_v(x|y)\|_2 \right] \leq \sqrt{2H_{d-1} \left( \mathcal{L}_{\exp}(\theta) - \mathcal{L}_{\exp}^\star \right)}. \tag{37}$$

$\square$

## C   ADDITIONAL RESULTS FOR *AutoGnothi*

In addition to the results on ImageNette included in the main paper, we provide more detailed results on other datasets ranging from image classification tasks to text classification tasks, against a number of baseline explanation methods, with respect to other model sizes in this section.

Table 4: Performance metrics for ViT-tiny on ImageNette.

| Method | Insertion (↑) | Deletion (↓) |
|---|---|---|
| Random | $0.9231_{\pm 0.1094}$ | $0.9229_{\pm 0.1106}$ |
| Attention last | $0.9281_{\pm 0.1033}$ | $0.7311_{\pm 0.2422}$ |
| Attention rollout | $0.9138_{\pm 0.1102}$ | $0.7306_{\pm 0.2449}$ |
| GradCAM (Attn) | $0.9155_{\pm 0.1242}$ | $0.8292_{\pm 0.2089}$ |
| GradCAM (LN) | $0.9280_{\pm 0.0937}$ | $0.8436_{\pm 0.2046}$ |
| Vanilla (Pixel) | $0.9006_{\pm 0.1173}$ | $0.8161_{\pm 0.2034}$ |
| Vanilla (Embed) | $0.9131_{\pm 0.1109}$ | $0.7708_{\pm 0.2272}$ |
| IntGrad (Pixel) | $0.9383_{\pm 0.0808}$ | $0.8734_{\pm 0.1817}$ |
| IntGrad (Embed) | $0.9317_{\pm 0.0808}$ | $0.8092_{\pm 0.1817}$ |
| SmoothGrad (Pixel) | $0.9153_{\pm 0.1199}$ | $0.7724_{\pm 0.2236}$ |
| SmoothGrad (Embed) | $0.9268_{\pm 0.1092}$ | $0.7852_{\pm 0.2176}$ |
| VarGrad (Pixel) | $0.9219_{\pm 0.1147}$ | $0.7872_{\pm 0.2157}$ |
| VarGrad (Embed) | $0.9317_{\pm 0.1063}$ | $0.8092_{\pm 0.2062}$ |
| LRP | $0.9439_{\pm 0.0852}$ | $0.6883_{\pm 0.2603}$ |
| Leave-one-out | $0.9632_{\pm 0.0401}$ | $0.7671_{\pm 0.2902}$ |
| RISE | $0.9743_{\pm 0.0333}$ | $0.6514_{\pm 0.3028}$ |
| Covert et al. | $\mathbf{0.9824_{\pm 0.0289}}$ | $0.5243_{\pm 0.2579}$ |
| *AutoGnothi* (Ours) | $0.9802_{\pm 0.0268}$ | $\mathbf{0.5097_{\pm 0.2688}}$ |

Table 5: Performance metrics for ViT-small on ImageNette.

| Method | Insertion (↑) | Deletion (↓) |
|---|---|---|
| Random | $0.9471_{\pm 0.0827}$ | $0.9461_{\pm 0.0843}$ |
| Attention last | $0.9599_{\pm 0.0771}$ | $0.7617_{\pm 0.2205}$ |
| Attention rollout | $0.9293_{\pm 0.0940}$ | $0.8566_{\pm 0.1793}$ |
| GradCAM (Attn) | $0.9217_{\pm 0.1205}$ | $0.9301_{\pm 0.1186}$ |
| GradCAM (LN) | $0.9239_{\pm 0.0893}$ | $0.9184_{\pm 0.1571}$ |
| Vanilla (Pixel) | $0.9535_{\pm 0.1006}$ | $0.8179_{\pm 0.1686}$ |
| Vanilla (Embed) | $0.9564_{\pm 0.0894}$ | $0.8240_{\pm 0.1886}$ |
| IntGrad (Pixel) | $0.9581_{\pm 0.0738}$ | $0.9219_{\pm 0.1281}$ |
| IntGrad (Embed) | $0.9581_{\pm 0.0738}$ | $0.9219_{\pm 0.1281}$ |
| SmoothGrad (Pixel) | $0.9542_{\pm 0.0770}$ | $0.7998_{\pm 0.2055}$ |
| SmoothGrad (Embed) | $0.9550_{\pm 0.0788}$ | $0.8065_{\pm 0.2090}$ |
| VarGrad (Pixel) | $0.9535_{\pm 0.0798}$ | $0.8179_{\pm 0.1954}$ |
| VarGrad (Embed) | $0.9564_{\pm 0.0776}$ | $0.8240_{\pm 0.1942}$ |
| LRP | $0.9636_{\pm 0.0637}$ | $0.7549_{\pm 0.2275}$ |
| Leave-one-out | $0.9684_{\pm 0.0319}$ | $0.8815_{\pm 0.2056}$ |
| RISE | $0.9773_{\pm 0.0206}$ | $0.7960_{\pm 0.2599}$ |
| Covert et al. | $\mathbf{0.9828_{\pm 0.0440}}$ | $0.6865_{\pm 0.2255}$ |
| *AutoGnothi* (Ours) | $0.9791_{\pm 0.0305}$ | $\mathbf{0.6667_{\pm 0.2636}}$ |

Table 6: Performance metrics for ViT-large on ImageNette.

| Method | Insertion ($\uparrow$) | Deletion ($\downarrow$) |
|---|---|---|
| Random | $0.9645_{\pm 0.0748}$ | $0.9642_{\pm 0.0757}$ |
| Attention last | $0.9251_{\pm 0.0794}$ | $0.8997_{\pm 0.1380}$ |
| Attention rollout | $0.9336_{\pm 0.0835}$ | $0.9429_{\pm 0.0997}$ |
| GradCAM (Attn) | $0.9398_{\pm 0.0692}$ | $0.9328_{\pm 0.1228}$ |
| GradCAM (LN) | $0.9590_{\pm 0.0619}$ | $0.9366_{\pm 0.1330}$ |
| Vanilla (Pixel) | $0.9040_{\pm 0.1046}$ | $0.9372_{\pm 0.1106}$ |
| Vanilla (Embed) | $0.9151_{\pm 0.0959}$ | $0.9258_{\pm 0.1237}$ |
| IntGrad (Pixel) | $0.9716_{\pm 0.0584}$ | $0.9596_{\pm 0.0948}$ |
| IntGrad (Embed) | $0.9716_{\pm 0.0584}$ | $0.9596_{\pm 0.0948}$ |
| SmoothGrad (Pixel) | $0.9499_{\pm 0.0778}$ | $0.8953_{\pm 0.1579}$ |
| SmoothGrad (Embed) | $0.9636_{\pm 0.0681}$ | $0.8664_{\pm 0.1643}$ |
| VarGrad (Pixel) | $0.9444_{\pm 0.0827}$ | $0.9060_{\pm 0.1456}$ |
| VarGrad (Embed) | $0.9558_{\pm 0.0687}$ | $0.8827_{\pm 0.1545}$ |
| LRP | $0.9506_{\pm 0.0646}$ | $0.8814_{\pm 0.1530}$ |
| Leave-one-out | $0.9743_{\pm 0.0521}$ | $0.9534_{\pm 0.1085}$ |
| RISE | $0.9801_{\pm 0.0373}$ | $0.9245_{\pm 0.1570}$ |
| Covert et al. | $\mathbf{0.9843_{\pm 0.0436}}$ | $0.7646_{\pm 0.2012}$ |
| *AutoGnothi* (Ours) | $0.9837_{\pm 0.0225}$ | $\mathbf{0.6570_{\pm 0.2171}}$ |

Table 7: Performance metrics for ViT-base on Oxford-IIIT Pet.

| Method | Insertion ($\uparrow$) | Deletion ($\downarrow$) |
|---|---|---|
| Random | $0.8642_{\pm 0.1855}$ | $0.8625_{\pm 0.1851}$ |
| Attention last | $0.9066_{\pm 0.1302}$ | $0.5534_{\pm 0.2183}$ |
| Attention rollout | $0.8616_{\pm 0.1384}$ | $0.7387_{\pm 0.2326}$ |
| GradCAM (Attn) | $0.8726_{\pm 0.1585}$ | $0.7582_{\pm 0.2296}$ |
| GradCAM (LN) | $0.8828_{\pm 0.1137}$ | $0.7648_{\pm 0.2462}$ |
| Vanilla (Pixel) | $0.8855_{\pm 0.1362}$ | $0.6551_{\pm 0.2395}$ |
| Vanilla (Embed) | $0.8996_{\pm 0.1354}$ | $0.5783_{\pm 0.2405}$ |
| IntGrad (Pixel) | $0.9219_{\pm 0.1137}$ | $0.8451_{\pm 0.1990}$ |
| IntGrad (Embed) | $0.9219_{\pm 0.1137}$ | $0.8451_{\pm 0.1990}$ |
| SmoothGrad (Pixel) | $0.9140_{\pm 0.1329}$ | $0.5508_{\pm 0.2347}$ |
| SmoothGrad (Embed) | $0.8731_{\pm 0.1462}$ | $0.8716_{\pm 0.1514}$ |
| VarGrad (Pixel) | $0.9145_{\pm 0.1268}$ | $0.5801_{\pm 0.2366}$ |
| VarGrad (Embed) | $0.8818_{\pm 0.1437}$ | $0.8778_{\pm 0.1487}$ |
| LRP | $0.9192_{\pm 0.1178}$ | $0.5362_{\pm 0.2258}$ |
| Leave-one-out | $0.9468_{\pm 0.0666}$ | $0.7341_{\pm 0.2951}$ |
| RISE | $\mathbf{0.9581_{\pm 0.0333}}$ | $0.6186_{\pm 0.3096}$ |
| Covert et al. | $0.9422_{\pm 0.1035}$ | $0.4958_{\pm 0.2404}$ |
| *AutoGnothi* (Ours) | $0.9384_{\pm 0.1088}$ | $\mathbf{0.4888_{\pm 0.2480}}$ |

Table 8: Performance metrics for ViT-base on MURA. MURA is a binary classification dataset, we calculated metrics for each of its two categories.

| Method | Abnormal | | Normal | |
|---|---|---|---|---|
| | Insertion ($\uparrow$) | Deletion ($\downarrow$) | Insertion ($\uparrow$) | Deletion ($\downarrow$) |
| Random | $0.8195_{\pm 0.1875}$ | $0.8206_{\pm 0.1859}$ | $0.1548_{\pm 0.1396}$ | $0.1564_{\pm 0.1415}$ |
| Attention last | $0.8416_{\pm 0.1863}$ | $0.6303_{\pm 0.1912}$ | $0.1546_{\pm 0.1365}$ | $0.1849_{\pm 0.1348}$ |
| Attention rollout | $0.8047_{\pm 0.1887}$ | $0.7236_{\pm 0.2107}$ | $0.1758_{\pm 0.1393}$ | $0.1636_{\pm 0.1366}$ |
| GradCAM (Attn) | $0.8077_{\pm 0.1883}$ | $0.8172_{\pm 0.1925}$ | $0.1611_{\pm 0.1387}$ | $0.1655_{\pm 0.1518}$ |
| GradCAM (LN) | $0.8509_{\pm 0.1787}$ | $0.7451_{\pm 0.2171}$ | $0.1771_{\pm 0.1537}$ | $0.1487_{\pm 0.1338}$ |
| Vanilla (Pixel) | $0.8384_{\pm 0.1764}$ | $0.5971_{\pm 0.2056}$ | $0.1710_{\pm 0.1401}$ | $0.1603_{\pm 0.1310}$ |
| Vanilla (Embed) | $0.8412_{\pm 0.1774}$ | $0.5709_{\pm 0.1993}$ | $0.1666_{\pm 0.1393}$ | $0.1649_{\pm 0.1295}$ |
| IntGrad (Pixel) | $0.8677_{\pm 0.1642}$ | $0.7690_{\pm 0.2261}$ | $0.2011_{\pm 0.1842}$ | $0.1326_{\pm 0.1283}$ |
| IntGrad (Embed) | $0.8677_{\pm 0.1642}$ | $0.7690_{\pm 0.2261}$ | $0.2011_{\pm 0.1842}$ | $0.1326_{\pm 0.1283}$ |
| SmoothGrad (Pixel) | $0.8351_{\pm 0.1893}$ | $0.6469_{\pm 0.2000}$ | $0.1610_{\pm 0.1428}$ | $0.1842_{\pm 0.1385}$ |
| SmoothGrad (Embed) | $0.8293_{\pm 0.1863}$ | $0.8006_{\pm 0.1971}$ | $0.1605_{\pm 0.1500}$ | $0.1552_{\pm 0.1406}$ |
| VarGrad (Pixel) | $0.8397_{\pm 0.1844}$ | $0.6667_{\pm 0.2012}$ | $0.1592_{\pm 0.1404}$ | $0.1813_{\pm 0.1453}$ |
| VarGrad (Embed) | $0.8328_{\pm 0.1841}$ | $0.8022_{\pm 0.1983}$ | $0.1575_{\pm 0.1461}$ | $0.1556_{\pm 0.1418}$ |
| LRP | $0.8524_{\pm 0.1786}$ | $0.6009_{\pm 0.1932}$ | $0.1693_{\pm 0.1459}$ | $0.1745_{\pm 0.1238}$ |
| Leave-one-out | $0.8996_{\pm 0.1336}$ | $0.6887_{\pm 0.2412}$ | $0.2952_{\pm 0.2235}$ | $0.0977_{\pm 0.0911}$ |
| RISE | $0.9247_{\pm 0.1037}$ | $0.6258_{\pm 0.2510}$ | $0.3470_{\pm 0.2431}$ | $0.0844_{\pm 0.0786}$ |
| Covert et al. | $\mathbf{0.9319}_{\pm \mathbf{0.0795}}$ | $0.4199_{\pm 0.2136}$ | $0.4516_{\pm 0.2506}$ | $\mathbf{0.0539}_{\pm \mathbf{0.0478}}$ |
| *AutoGnothi* (Ours) | $0.9292_{\pm 0.0597}$ | $\mathbf{0.4116}_{\pm \mathbf{0.2116}}$ | $\mathbf{0.4563}_{\pm \mathbf{0.2524}}$ | $0.0581_{\pm 0.0488}$ |

Table 9: Performance metrics for Bert-base on Yelp Review Polarity.

| Method | Insertion ($\uparrow$) | Deletion ($\downarrow$) |
|---|---|---|
| KernelShap | $0.8894_{\pm 0.1324}$ | $0.4624_{\pm 0.2548}$ |
| Covert et al. | $\mathbf{0.9620}_{\pm \mathbf{0.0472}}$ | $0.1725_{\pm 0.1176}$ |
| *AutoGnothi* (Ours) | $0.9588_{\pm 0.0206}$ | $\mathbf{0.1004}_{\pm \mathbf{0.0377}}$ |

# D ABLATION STUDY ON THE SIZE OF SIDE-NETWORKS

In this section, we present an ablation study on the architecture and size of side-networks in *Auto-Gnothi*. Specifically, we investigated the effect of varying the reduction factor ($r = 2, 4, 8, 16, 32$) on the side-networks of ViT-base model. To evaluate the impact, we compared the performance of *AutoGnothi* with Covert et al. (Covert et al., 2022) on the ImageNette dataset. The evaluation was performed using the insertion and deletion metrics, and the results are summarized in Table 10.

This study aimed to assess the balance between computational efficiency and explanation quality under different configurations of the side-network. The findings are as follows:

- **Too Large** $r$**:** When the reduction factor $r$ was too large (resulting in a very small side-network), the network lacked sufficient capacity to learn the explanations effectively. As a result, the explanation quality degraded, with the side-network failing to capture enough information from the main branch.

- **Too Small** $r$**:** Conversely, when $r$ was too small (resulting in a larger side-network), the network consumed excessive computational resources and interfered with the shared features of the predictor. This interference reduced the feature similarity between the prediction and explanation tasks, negatively impacting the faithfulness of the explainer.

- **Moderate** $r$**:** A moderate reduction factor (*e.g.*, $r = 8$) provided the optimal trade-off between computational efficiency and explanation quality. This configuration allowed the side-network to effectively utilize the features from the predictor while maintaining resource efficiency.

These results highlight the importance of carefully selecting the reduction factor to optimize the performance of *AutoGnothi* across different transformer architectures. The identified trade-offs ensure that the side-network remains both effective and efficient, making it suitable for various applications.

Table 10: Explanation Quality Metrics of the ViT-base Explainer on ImageNette dataset.

| Method | Insertion ($\uparrow$) | Deletion ($\downarrow$) |
|---|---|---|
| Covert et al. | $0.9839_{\pm 0.0375}$ | $0.8121_{\pm 0.1768}$ |
| *AutoGnothi* ($r$=2) | $0.9894_{\pm 0.0329}$ | $0.8687_{\pm 0.1806}$ |
| *AutoGnothi* ($r$=4) | $0.9857_{\pm 0.0251}$ | $0.7846_{\pm 0.1957}$ |
| *AutoGnothi* ($r$=8) | $0.9874_{\pm 0.0265}$ | $0.7954_{\pm 0.2294}$ |
| *AutoGnothi* ($r$=16) | $0.9720_{\pm 0.0288}$ | $0.6202_{\pm 0.2023}$ |
| *AutoGnothi* ($r$=32) | $0.9520_{\pm 0.0443}$ | $0.5547_{\pm 0.1941}$ |

# E    ADDITIONAL VISUALIZATIONS FOR *AutoGnothi*

In this section we present a number of image samples on the ViT-base model, regarding explanation outputs from 12 representative baseline explanation methods. We ensure that these samples correspond to correct model predictions made by the base model to ensure better clarity.

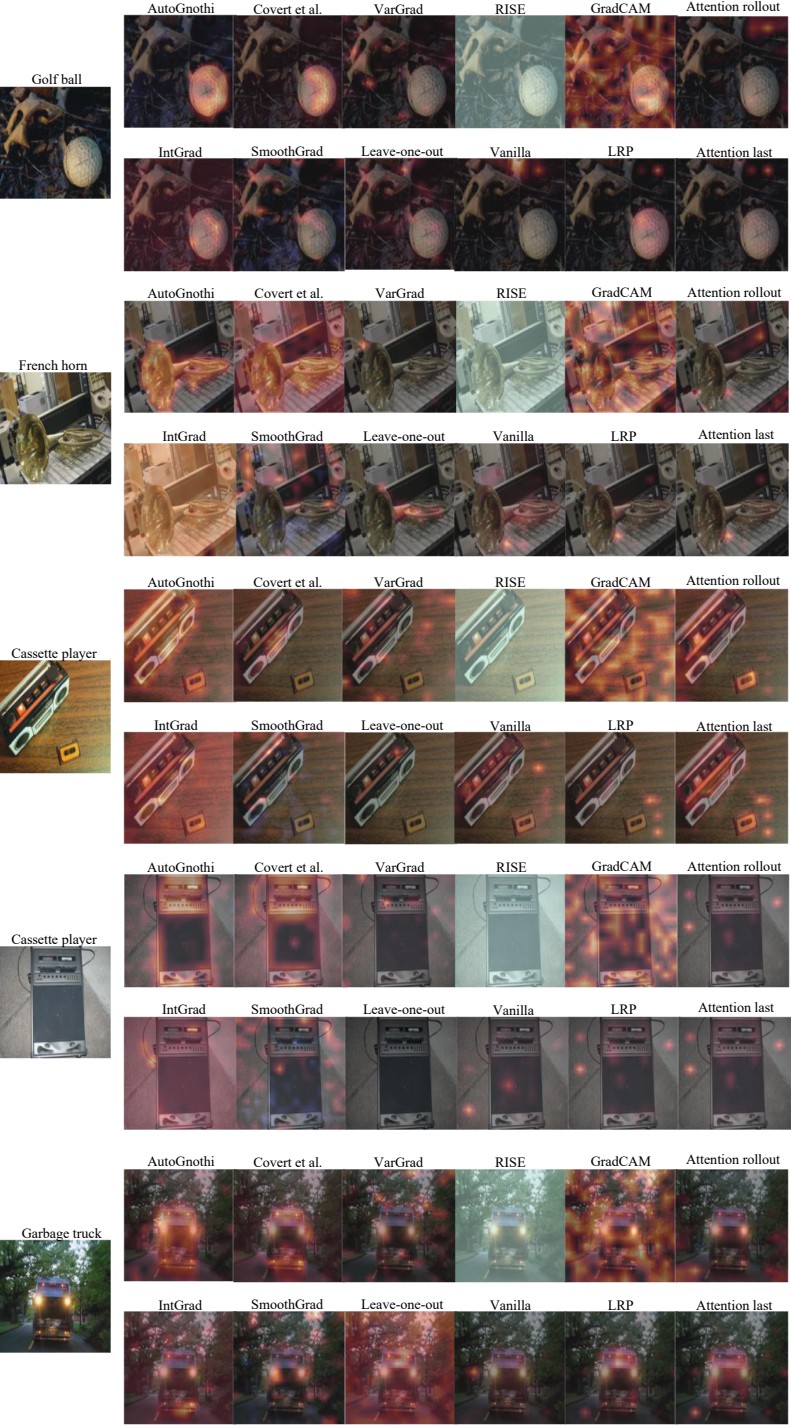

Figure 8: Visualization of ViT-base explanation on ImageNette dataset (1/2).

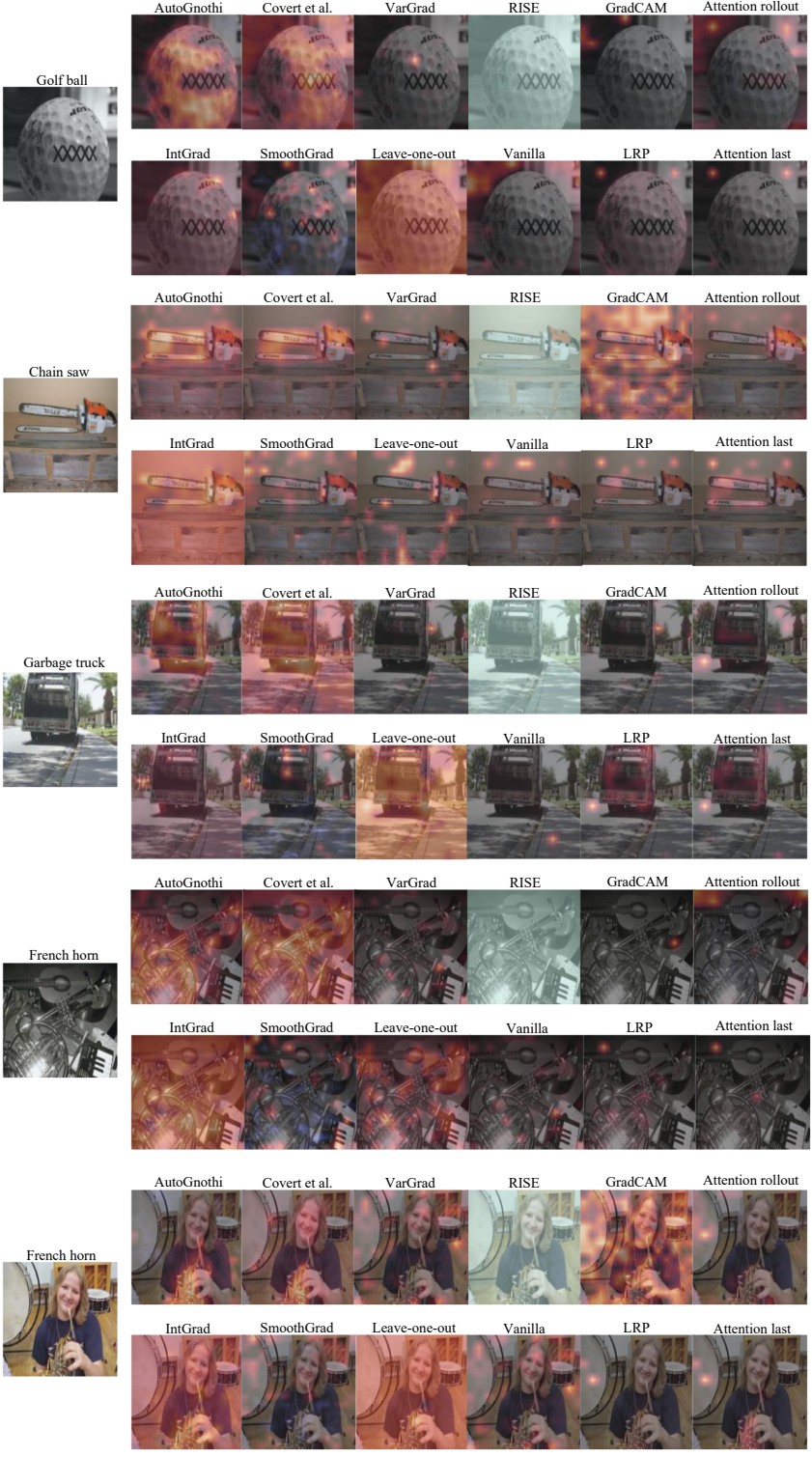

Figure 9: Visualization of ViT-base explanation on ImageNette dataset (2/2).

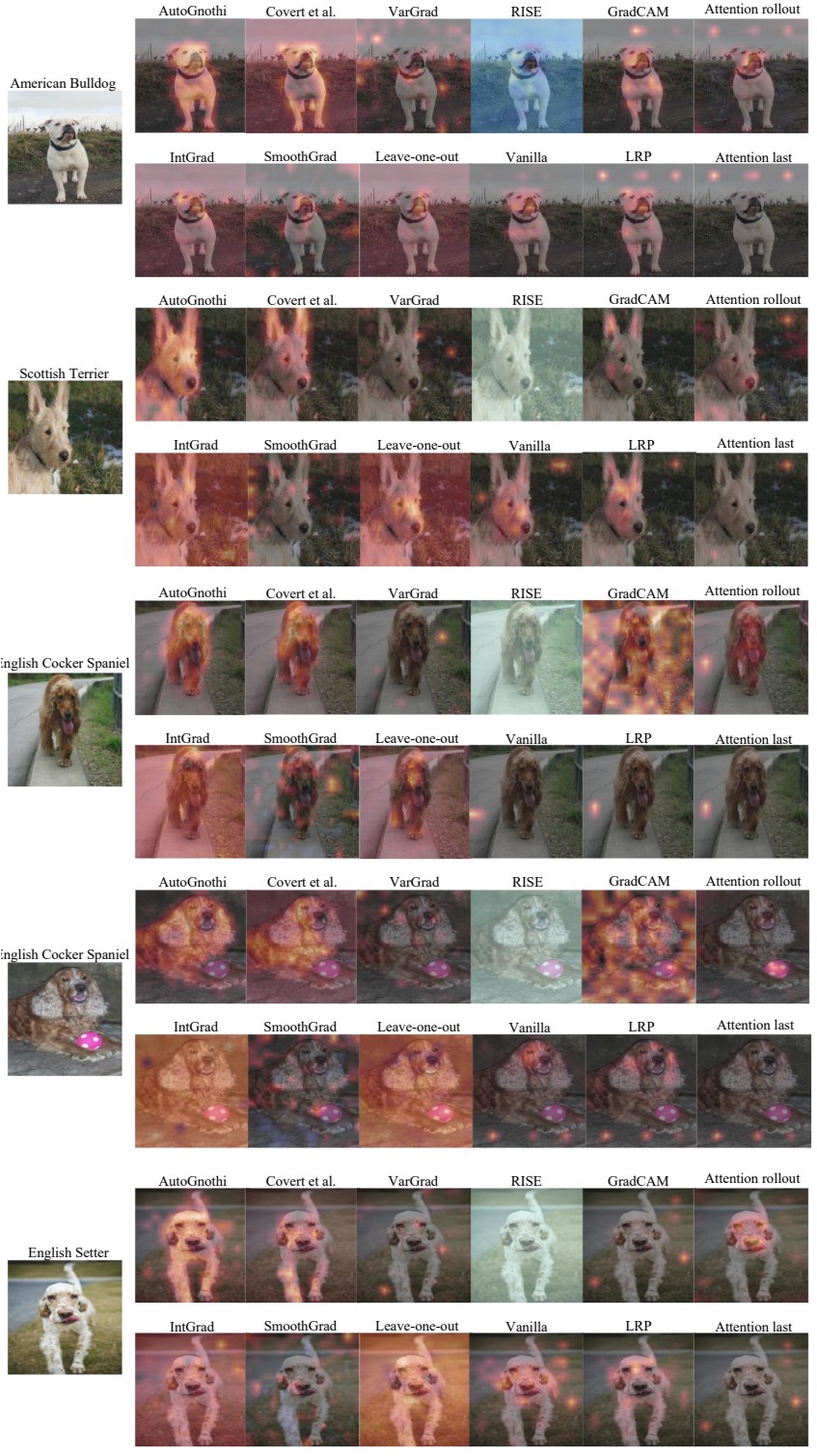

Figure 10: Visualization of ViT-base explanation on Oxford-IIIT Pet dataset (1/2).

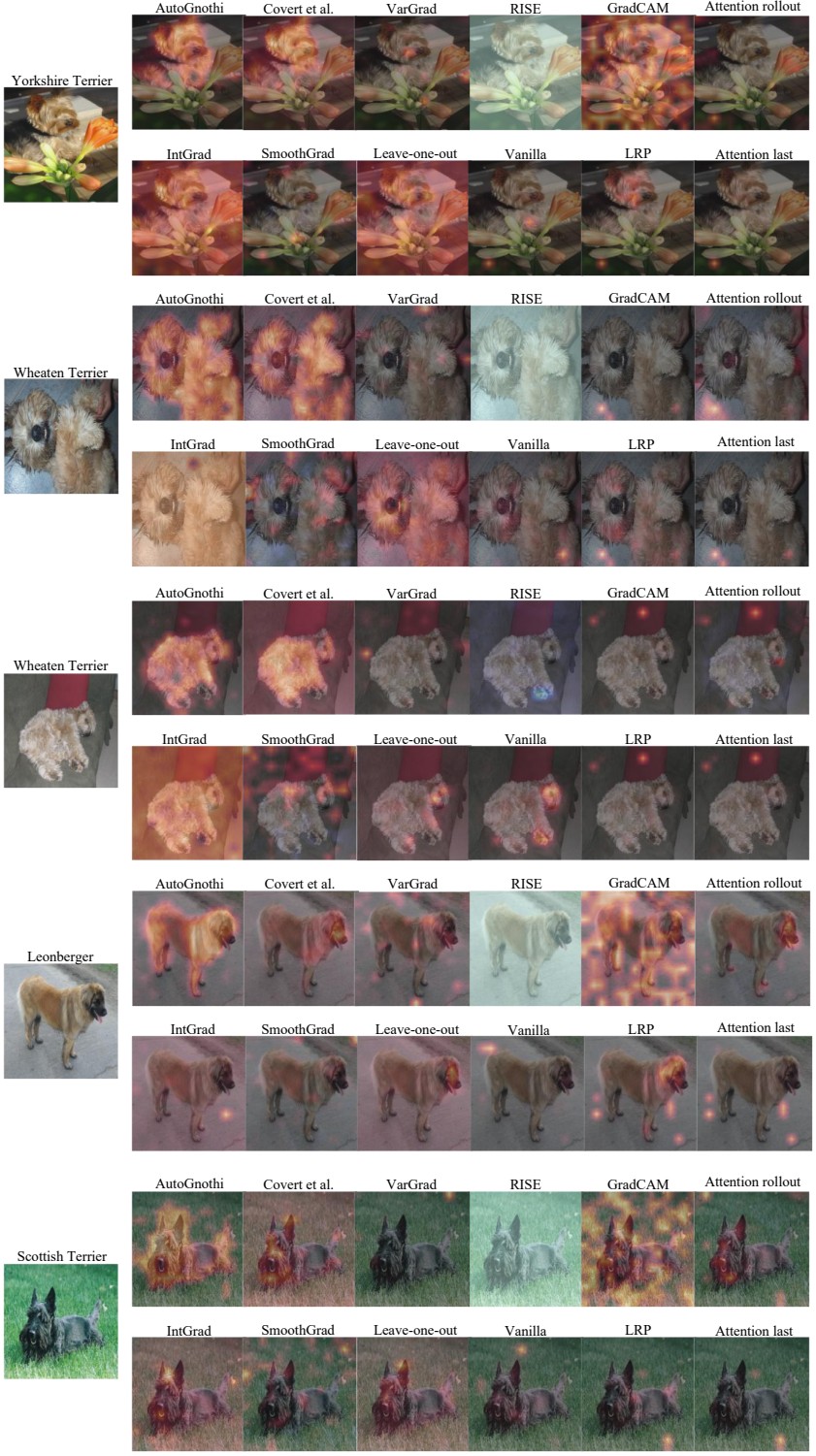

Figure 11: Visualization of ViT-base explanation on Oxford-IIIT Pet dataset (2/2).

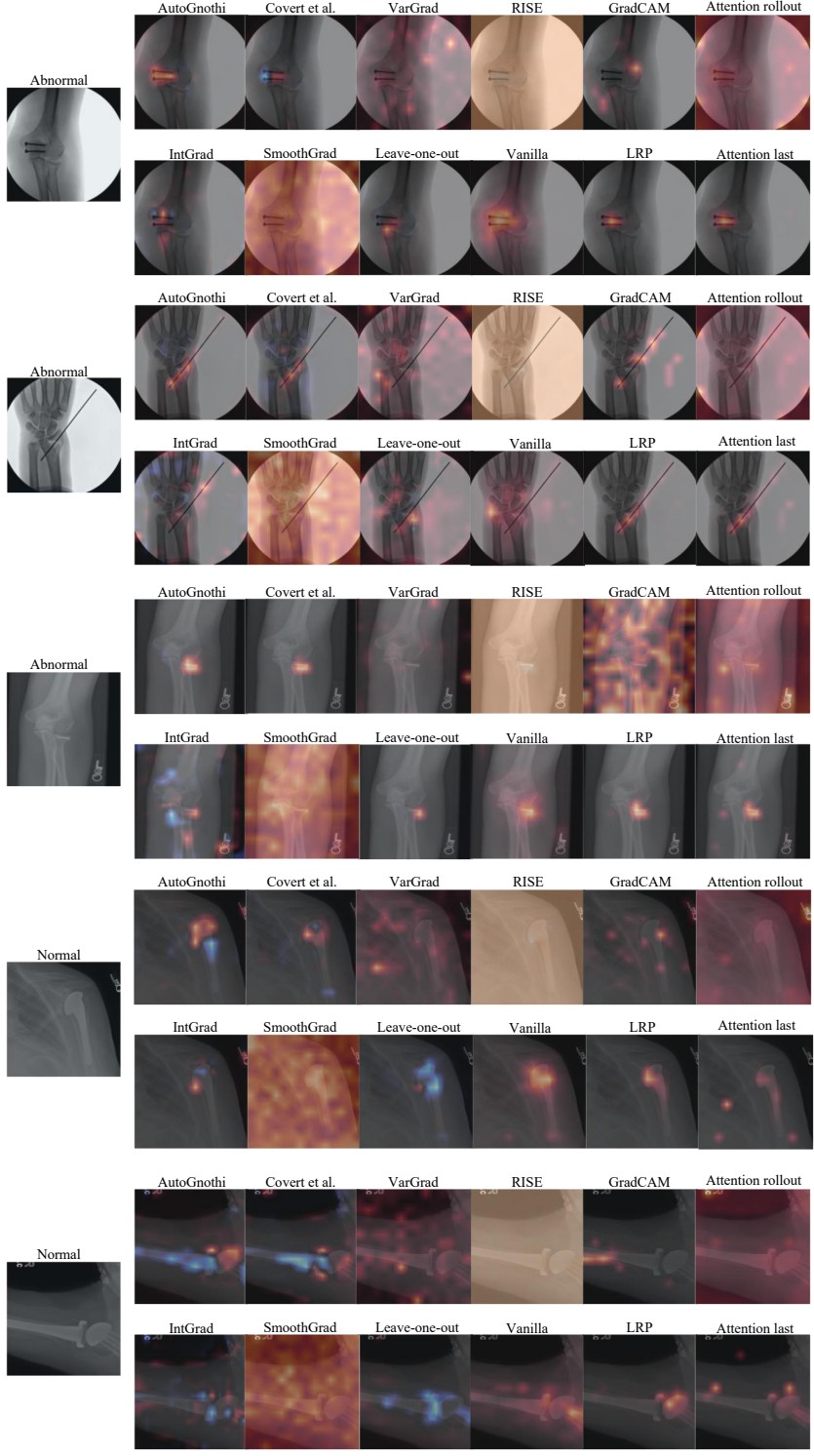

Figure 12: Visualization of ViT-base explanation on MURA dataset (1/2).

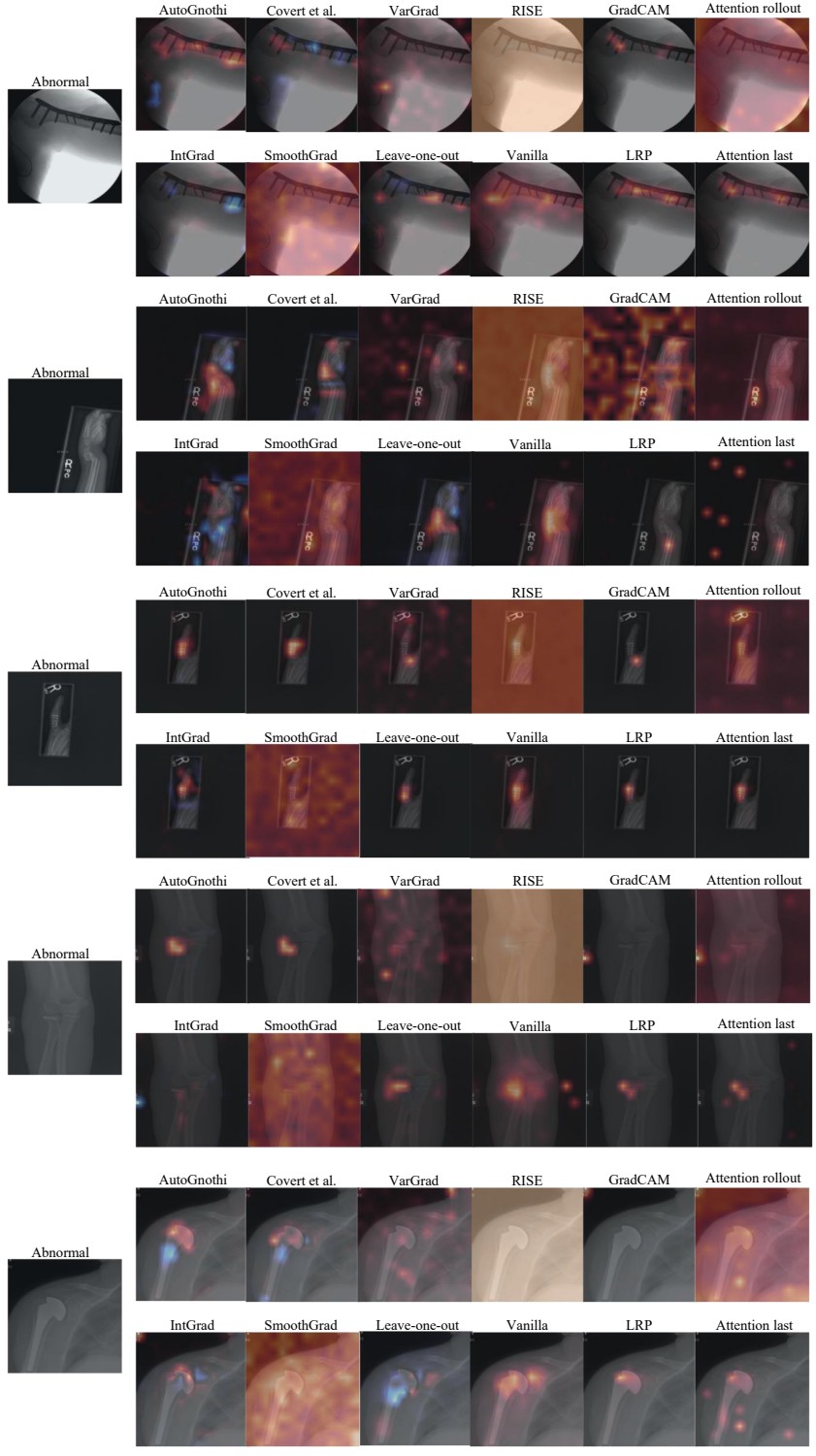

Figure 13: Visualization of ViT-base explanation on MURA dataset (2/2).

