# OpenReview forum: "Gnothi Seauton: Empowering Faithful Self-Interpretability in Black-Box Transformers"
_ICLR.cc/2025/Conference — ICLR 2025 Poster_

### Official Review · Reviewer_2vJv · 2024-10-21

**Soundness:** 3
**Presentation:** 4
**Contribution:** 2
**Rating:** 6
**Confidence:** 4

**Summary:**

This paper is concerned with the setting of training an explainer for an explainee.
Improvements to the previous work, which trains a surrogate model that can handle masked inputs and another explainer, that can predict the saliency, are primarily in terms of required ressources, both for training and inference with respect to time and memory.
This is achieved by utilizing Ladder-Side-Tuning to only train a reduced number of parameters on top of the explainee.
When saliency and prediction are both desired, only the explainee and the few additional parameters are needed, incurring significantly reduced costs, while still matching the previous SOTA performance.

**Strengths:**

The paper is presented very clearly, well written and easy to understand.

The proposed method combines ideas of PETL and Covert et al. in an original way.

The method at least matches the performance of the significantly more resource intensive method of Covert et al., clearly improving upon it overall.

The evaluation covers two modalities and multiple datasets with consistent results.

A convincing ablation demonstrates the superiority of the proposed approach to two naive baselines, called Froyo and Duo, showing that explaining and predicting cause orthogonal gradients.

**Weaknesses:**

The paper discusses a  simple, resource efficient alternative to a niche paradigm of explaining a model, that is only applicable to Transformers.

Patch Dropout has recently emerged as training augmentation for Vision Transformers, that enables transformers to handle masked inputs. (https://openaccess.thecvf.com/content/WACV2023/papers/Liu_PatchDropout_Economizing_Vision_Transformers_Using_Patch_Dropout_WACV_2023_paper.pdf)
Since the surrogate models are more accurate on multiple datasets, it is not clear why one would want to stick to the explainee.
The paper misses a discussion of that.

Combining the two points above, I am unsure how significant this work is, as the field might move towards Transformers trained with Patch Dropout.

While the results in Table 3 suggest a gap, they are cherry-picked and no "clear margin" is visible to Covert et al. across the different architectures, typcially more evenly matched (As in Tab.  4 -8).

The evaluation using the surrogate model is frickle, as it is not actually the model to be explained.

It is unclear why ImageNette is used as main dataset, as it seems too easy, with accuracies above 99%.

Fig. 2 is too full. The table is repeated in the paper anyway. (Table 1 and 2)

typos in lines : 264, 535

**Questions:**

How do the saliency maps transfer across surrogate models?
As the saliency maps explain the explainee, they should work across a range of surrogate models, especially those the explainers were not trained on,  but which still try to explain the same explainee.

Which surrogate model is used to evaluate the baselines?

---

> ### Comment · Reviewer_2vJv · 2024-11-28
> **Official Comment by Reviewer 2vJv**
>
> Thank you for this extensive discussion and providing these very interesting results.\
> As it stands, I still think that the main evaluation pipeline followed in this paper is flawed, but the new results evaluated across different surrogates seem to show at least similar performance to Covert et al.\
> Because I like the general idea and the clear drastic improvements compared to ViT-Shapley in terms of efficiency, I will raise my score despite the holistically suboptimal evaluation.
>
> I would encourage the authors to include some aspects of this discussion into the paper.

---

### Official Review · Reviewer_Qio4 · 2024-11-02

**Soundness:** 3
**Presentation:** 4
**Contribution:** 3
**Rating:** 6
**Confidence:** 5

**Summary:**

The paper proposes a method called AutoGnothi for generating Shapley-based explanations in black-box models, such as Vision Transformers (ViTs) and BERT, using side-tuning. Instead of modifying the main model, AutoGnothi trains a lightweight, parallel "surrogate" network that learns to approximate the importance of different input features without requiring iterative masking during inference, as is typically needed for Shapley values.
The main advantage of AutoGnothi lies in its efficiency. By using side-tuning, AutoGnothi is more memory- and time-efficient compared to previous approaches like ViT-Shapley, which require a fully fine-tuned surrogate model.

The authors measure the quality of explanations using insertion and deletion metrics, which test whether removing or adding key features impacts the model’s prediction as expected. In these tests, AutoGnothi’s explanations are shown to be comparable to ViT-Shapley (I read through the Appendix as well to make this conclusion). Besides, although qualitative examples suggest that AutoGnothi may better highlight main objects in images, the paper doesn’t fully convince me that it strictly produces better explanations than ViT-Shapley. Putting the explanations into a human-in-the-loop evaluation would help answer my concern.

The general idea is not new (e.g. compared vs. ViT-Shapley) but I appreciate the efficient solution, clear/detailed writing style, and thorough evaluation. I especially enjoyed reading Section 4.2.

In summary, I like the paper and recommend accepting it. My ratings can be adjusted during the rebuttal.

**Strengths:**

Originality: While AutoGnothi's goal of providing Shapley-based explanations overlaps with previous work like ViT-Shapley, it introduces a new side-tuning approach to reduce memory and computational demands.

Quality: The paper provides a thorough evaluation across multiple datasets (both vision and language), using various metrics to prove the efficiency and explanation faithfulness.

Clarity: The writing is clear and  well-organized.

**Weaknesses:**

The evaluation of explanation:
Although qualitative examples (Fig.7) suggest that AutoGnothi may better highlight main objects in images, the paper doesn’t fully convince me that it strictly produces better explanations than ViT-Shapley. Putting the explanations into a human-in-the-loop evaluation [1,2] would strengthen claims about its explanation quality.

[1] The Effectiveness of Feature Attribution Methods and Its Correlation with Automatic Evaluation Scores

[2] What I Cannot Predict, I Do Not Understand: A Human-Centered Evaluation Framework for Explainability Methods

Minor: I think the authors could also be upfront about the limitations of the works (e.g. approximation, explanation evaluation). The current writing does not discuss any of its **possible** limitations.

**Questions:**

Q1. The general idea in AutoGnothi is that the main model makes predictions, and the surrogate model explains those predictions. However, how can the authors ensure that the surrogate model **truly** reflects how the main model makes its decisions? Although the paper shows efforts to align the two main and surrogate models, how well would AutoGnothi work if we scale it to more complex models (where aligning networks structure is prohibited) with complicated output distributions, where using KL divergence might not be enough to keep them aligned? Are there better alternatives than KL divergence?

Q2. I’m curious to see how AutoGnothi performs on more complex tasks. Currently, the tasks feel easy (especially the text classification). I’m concerned that side-tuning may not scale well with more complex tasks, like question answering or fine-grained image classification (e.g., CUB-200). How would the authors think the current side-tuning can satisfy interpretability and accuracy for complex tasks?

To clarify more, Q1 is about alignment and Q2 is about the feasibility of side-tuning.

---

### Official Review · Reviewer_5F4V · 2024-11-04

**Soundness:** 3
**Presentation:** 4
**Contribution:** 3
**Rating:** 6
**Confidence:** 4

**Summary:**

This paper aims to address the tradeoff between prediction accuracy and interpretability of machine learning models. That is develop an inherently explainable model without compromising its prediction performance. To achieve this, the authors propose to integrate small side networks into black-box models and use them to generate Shapley value explanations. The paper provides comparisons of computational costs and memory usage, and empirical results on both prediction accuracy and explanation quality.

**Strengths:**

1. The idea is intuitive and novel.
2. The paper is well-written and easy to follow.
3. The proposed method significantly cuts down the trainable parameters, memory usage, and FLOPs without compromising prediction accuracy and explanation quality.

**Weaknesses:**

1. Since the proposed method relies on the frozen pretrained black-box model’s parameters, will the biases or spurious correlations in the pretrained model also affect the generated explanations from the side network? Can the side network mitigate these issues?
2. Following Weakness 1, robustness evaluations are missing from the current experiments. How the side network will behave under the out-of-distribution (OoD) or even adversarial data w.r.t. the pretrained model?
3. The side network is designed to be parameter-efficient, but is there an optimal balance between its complexity and explanation fidelity? It would be better to see ablation studies on different side network sizes or architectures.

**Questions:**

See Weaknesses for details.

---

### Author Response · Authors · 2024-12-03
**Thanks to All Reviewers for Their Thoughtful Feedback**

We sincerely thank all the reviewers for their thoughtful feedback and valuable suggestions during the rebuttal process. Their efforts have greatly contributed to improving the quality of our paper by inspiring additional experiments under new settings and fostering deeper investigations.

**Reviewer 5F4V** rated the paper a **6**, acknowledging that "The proposed method **significantly cuts down the trainable parameters, memory usage, and FLOPs** without compromising prediction accuracy and explanation quality." This aligns with the main contribution of our work, which seeks to achieve efficient and faithful self-interpretability for black-box models.

**Reviewer Qio4** also rated the paper a **6**, increasing the confidence score from **3 to 5**. The reviewer stated: "I will increase my confidence score from 3 to 5 to reflect my growing support for **accepting** this paper." After the rebuttal, the reviewer confirmed, "I am maintaining my current rating and lean towards **accepting** this paper."

**Reviewer 2vJv** raised the score from **5 to 6** after the rebuttal. The reviewer provided valuable suggestions regarding evaluation with multiple surrogates and further modifications of the pipeline with PatchDropout, offering insights into the current two-step training paradigm of our work and related methods like ViT-Shapley. These concerns were successfully addressed during the rebuttal. Reviewer 2vJv noted: "Because I like the general idea and the clear drastic improvements compared to ViT-Shapley in terms of efficiency, I will raise my score."

We are thrilled that **all three reviewers unanimously agree this paper should be _accepted_**. Moving forward, we will continue refining this work by further optimizing the experiments and details introduced during the rebuttal process and integrating them into the final version of the paper. Thank you again for your time and thoughtful efforts.

---

### Meta-Review · Area_Chair_9D7Q · 2024-12-18

**Metareview:**

This paper addresses the tradeoff between prediction accuracy and interpretability in machine learning models by proposing an approach to integrate small side networks into black-box models. These side networks generate Shapley value explanations, aiming to achieve inherent explainability without compromising predictive performance. The study includes comparisons of computational costs, memory usage, and empirical results on both prediction accuracy and explanation quality. Reviewers scored the paper 6, acknowledging its novelty and value. However, they also provided suggestions for improvement, which the authors should address in the next version. Overall, I recommend accepting this paper.

**Additional Comments On Reviewer Discussion:**

. Reviewer  5F4V raised concerns about whether biases or spurious correlations in the pretrained model might affect the generated explanations, the robustness of the explainer to out-of-distribution (OoD) data, and the optimal complexity of the side network. The authors addressed these concerns by providing additional experiments and clarifications to resolve the issues.

Reviewer Qio4 highlighted weaknesses, including the need for a Human-In-The-Loop evaluation of explanation quality, improved loss functions, surrogate model evaluation, and performance on complex tasks. The authors addressed these issues through detailed explanations and new experimental results, effectively resolving the concerns.


Reviewer 2vJv expressed concerns about the evaluation of the surrogate model and the need for more results on additional datasets. The authors addressed these concerns by conducting new experiments and providing additional results to strengthen their findings.


All mentioned paper novelty and value for the field

---

### Decision · Program_Chairs · 2025-01-22

Accept (Poster)